# Zero Time Waste: Recycling Predictions in Early Exit Neural Networks

**Maciej Wołczyk**[*][†]
Jagiellonian University

**Bartosz Wójcik**[*]
Jagiellonian University

**Klaudia Bałazy**
Jagiellonian University

**Igor Podolak**
Jagiellonian University

**Jacek Tabor**
Jagiellonian University

**Marek Śmieja**
Jagiellonian University

**Tomasz Trzciński**
Jagiellonian University,
Warsaw University of Technology,
Tooploox

## Abstract

The problem of reducing processing time of large deep learning models is a fundamental challenge in many real-world applications. Early exit methods strive towards this goal by attaching additional Internal Classifiers (ICs) to intermediate layers of a neural network. ICs can quickly return predictions for easy examples and, as a result, reduce the average inference time of the whole model. However, if a particular IC does not decide to return an answer early, its predictions are discarded, with its computations effectively being wasted. To solve this issue, we introduce Zero Time Waste (ZTW), a novel approach in which each IC reuses predictions returned by its predecessors by (1) adding direct connections between ICs and (2) combining previous outputs in an ensemble-like manner. We conduct extensive experiments across various datasets and architectures to demonstrate that ZTW achieves a significantly better accuracy vs. inference time trade-off than other recently proposed early exit methods.

## 1 Introduction

Deep learning models achieve tremendous successes across a multitude of tasks, yet their training and inference often yield high computational costs and long processing times [11, 22]. For some applications, however, efficiency remains a critical challenge, *e.g.* to deploy a reinforcement learning (RL) system in production the policy inference must be done in real-time [7], while the robot performances suffer from the delay between measuring a system state and acting upon it [34]. Similarly, long inference latency in autonomous cars could impact its ability to control the speed [13] and lead to accidents [10, 17].

Typical approaches to reducing the processing complexity of neural networks in latency-critical applications include compressing the model [24, 26, 46] or approximating its responses [21]. For instance, Livne & Cohen [26] propose to compress a RL model by policy pruning, while Kouris et al. [21] approximate the responses of LSTM-based modules in self-driving cars to accelerate their inference time. While those methods improve processing efficiency, they still require samples to pass

---

[*]equal contribution

[†]Corresponding author: `maciej.wolczyk@doctoral.uj.edu.pl`

35th Conference on Neural Information Processing Systems (NeurIPS 2021).

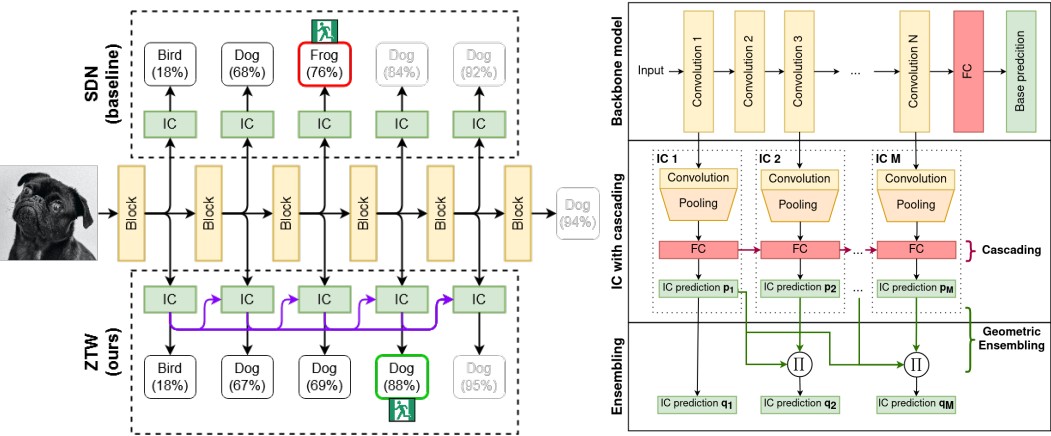

(a) Comparison of the proposed ZTW (bottom) with a conventional early-exit model, SDN (top).

(b) Detailed scheme of the proposed ZTW model architecture.

Figure 1: (a) In both approaches, internal classifiers (ICs) attached to the intermediate hidden layers of the base network allow us to return predictions quickly for examples that are easy to process. While SDN discards predictions of uncertain ICs (*e.g.* below a threshold of 75%), ZTW reuses computations from all previous ICs, which prevents information loss and waste of computational resources. (b) Backbone network $f_\theta$ lends its hidden layer activations to ICs, which share inferred information using cascade connections (red horizontal arrows in the middle row) and give predictions $p_m$. The inferred predictions are combined using ensembling (bottom row) giving $q_m$.

through the entire model. In contrast, biological neural networks leverage simple heuristics to speed up decision making, *e.g.* by shortening the processing path even in case of complex tasks [1, 9, 18].

This observation led a way to the inception of the so-called *early exit* methods, such as Shallow-Deep Networks (SDN) [19] and Patience-based Early Exit (PBEE) [47], that attach simple classification heads, called internal classifiers (ICs), to selected hidden layers of neural models to shorten the processing time. If the prediction confidence of a given IC is sufficiently high, the response is returned, otherwise, the example is passed to the subsequent classifier. Although these models achieve promising results, they discard the response returned by early ICs in the evaluation of the next IC, disregarding potentially valuable information, e.g. decision confidence, and wasting computational effort already incurred.

Motivated by the above observation, we postulate to look at the problem of neural model processing efficiency from the *information recycling* perspective and introduce a new family of *zero waste models*. More specifically, we investigate how information available at different layers of neural models can contribute to the decision process of the entire model. To that end, we propose Zero Time Waste (ZTW), a method for an intelligent aggregation of the information from previous ICs. A high-level view of our model is given in Figure 1. Our approach relies on combining ideas from networks with skip connections [41], gradient boosting [3], and ensemble learning [8, 23]. Skip connections between subsequent ICs (which we call *cascade connections*) allow us to explicitly pass the information contained within low-level features to the deeper classifier, which forms a cascading structure of ICs. In consequence, each IC improves on the prediction of previous ICs, as in gradient boosting, instead of generating them from scratch. To give the opportunity for every IC to explicitly reuse predictions of all previous ICs, we additionally build an ensemble of shallow ICs.

We evaluate our approach on standard classification benchmarks, such as CIFAR-100 and ImageNet, as well as on the more latency-critical applications, such as reinforcement-learned models for interacting with sequential environments. To the best of our knowledge, we are the first to show that early exit methods can be used for cutting computational waste in a reinforcement learning setting.

Results show that ZTW is able to save much more computation while preserving accuracy than current state-of-the-art early exit methods. In order to better understand where the improvements come from, we introduce Hindsight Improvability, a metric for measuring how efficiently the model

reuses information from the past. We provide ablation studies and additional analysis of the proposed method in the Appendix.

To summarize, the contributions of our work are the following:

- We introduce a family of zero waste models that quantify neural network efficiency with the Hindsight Improvability metrics.
- We propose an instance of zero waste models dubbed Zero Time Waste (ZTW) method which uses cascade connections and ensembling to reuse the responses of previous ICs for the final decision.
- We show how the state-of-the-art performance of ZTW in the supervised learning scenario generalizes to reinforcement learning.

## 2 Related Work

The drive towards reducing computational waste in deep learning literature has so far focused on reducing the inference time. Numerous approaches for accelerating deep learning models focus on building more efficient architectures [15], reducing the number of parameters [12] or distilling knowledge to smaller networks [14]. Thus, they decrease inference time by reducing the overall complexity of the model instead of using the conditional computation framework of adapting computational effort to each example. As such we find them orthogonal to the main ideas of our work, e.g. we show that applying our method to architectures designed for efficiency, such as MobileNet [15], leads to even further acceleration. Hence, we focus here on methods that adaptively set the inference time for each example.

**Conditional Computation**    Conditional computation was first proposed for deep neural networks in Bengio et al. [2] and Davis & Arel [5], and since then many sophisticated methods have been proposed in this field, including dynamic routing [27], cascading with multiple networks [40] and skipping intermediate layers [41] or channels [42]. In this work, we focus on the family of early exit approaches, as they usually do not require special assumptions about the underlying architecture of the network and the training paradigm, and because of that can be easily applied to many commonly used architectures. In BranchyNet [38] a loss function consisting of a weighted sum of individual head losses is utilized in training, and entropy of the head prediction is used for the early exit criterion. Berestizshevsky & Guy [4] propose to use confidence (maximum of the softmax output) instead. A broader overview of early exit methods is available in Scardapane et al. [32].

Several works proposed specialized architectures for conditional computation which allow for multi-scale feature processing [16, 44, 43], and developed techniques to train them more efficiently by passing information through the network [29, 25]. However, in this paper, we consider the case of increasing inference speed of a pre-trained network based on an architecture which was not built with conditional computation or even efficiency in mind. We argue that this is a practical use case, as this approach can be used to a wider array of models. As such, we do not compare with these methods directly.

Shallow-Deep Networks (SDN) [19] is a conceptually simple yet effective method, where the comparison of confidence with a fixed threshold is used as the exit criterion. The authors attach internal classifiers to layers selected based on the number of compute operations needed to reach them. The answer of each head is independent of the answers of the previous heads, although in a separate experiment the authors analyze the measure of disagreement between the predictions of final and intermediate heads.

Zhou et al. [47] propose Patience-based Early Exit (PBEE) method, which terminates inference after $t$ consecutive unchanged answers, and show that it outperforms SDN on a range of NLP tasks. The idea of checking for agreement in preceding ICs is connected to our approach of reusing information from the past. However, we find that applying PBEE in our setting does not always work better than SDN. Additionally, in the experiments from the original work, PBEE was trained simultaneously along with the base network, thus making it impossible to preserve the original pre-trained model.

**Ensembles**    Ensembling is typically used to improve the accuracy of machine learning models [6]. Lakshminarayanan et al. [23] showed that it also greatly improves calibration of deep neural networks. There were several attempts to create an ensemble from different layers of a network. Scardapane et

al. [31] adaptively exploit outputs of all internal classifiers, albeit not in a conditional computation context. Phuong & Lampert [29] used averaged answers of heads up to the current head for anytime-prediction, where the computational budget is unknown. Besides the method being much more basic, their setup is notably different from ours, as it assumes the same computational budget for all samples no matter how difficult the example is. Finally, none of the ensemble methods mentioned above were designed to work with pre-trained models.

## 3 Zero Time Waste

Our goal is to reduce computational costs of neural networks by minimizing redundant operations and information loss. To achieve it, we use the conditional computation setting, in which we dynamically select the route of an input example in a neural network. By controlling the computational route, we can decide how the information is stored and utilized within the model for each particular example. Intuitively, difficult examples require more resources to process, but using the same amount of compute for easy examples is wasteful. Below we describe our Zero Time Waste method in detail.

In order to adapt already trained models to conditional computation setting, we attach and train early exit classifier heads on top of several selected layers, without changing the parameters of the base network. During inference, the whole model exits through one of them when the response is likely enough, thus saving computational resources.

Formally, we consider a multi-class classification problem, where $x \in \mathbb{R}^D$ denotes an input example and $y \in \{1, \ldots, K\}$ is its target class. Let $f_\theta : \mathbb{R}^D \to \mathbb{R}^K$ be a pre-trained neural network with logit output designed for solving the above classification task. The weights $\theta$ will not be modified.

**Model overview**   Following typical early exit frameworks, we add $M$ shallow Internal Classifiers, $\text{IC}_1, \ldots, \text{IC}_M$, on intermediate layers of $f_\theta$. Namely, let $g_{\phi_m}$, for $m \in \{1, \ldots, M\}$, be the $m$-th IC network returning $K$ logits, which is attached to hidden layer $f_{\theta_m}$ of the base network $f_\theta$. The index $m$ is independent of $f_\theta$ layer numbering. In general, $M$ is lower than the overall number of $f_\theta$ hidden layers since we do not add ICs after every layer (see more details in Appendix A.1).

Although using ICs to return an answer early can reduce overall computation time [19], in a standard setting each IC makes its decision independently, ignoring the responses returned by previous ICs. As we show in Section 4.2, early layers often give correct answers for examples that are misclassified by later classifiers, and hence discarding their information leads to waste and performance drops. To address this issue, we need mechanisms that collect the information from the first $(m-1)$ ICs to inform the decision of $\text{IC}_m$. For this purpose, we introduce two complementary techniques: *cascade connections* and *ensembling*, and show how they help reduce information waste and, in turn, accelerate the model.

Cascade connections directly transfer the already inferred information between consecutive ICs instead of re-computing it again. Thus, they improve the performance of initial ICs that lack enough predictive power to classify correctly based on low-level features. Ensembling of individual ICs improves performance as the number of members increases, thus showing greatest improvements in the deeper part of the network. This is visualized in Figure 1 where cascade connections are used first to pass already inferred information to later ICs, while ensembling is utilized to conclude the IC prediction. The details on these two techniques are presented in the following paragraphs.

**Cascade connections**   Inspired by the gradient boosting algorithm and literature on cascading classifiers [39], we allow each IC to improve on the predictions of previous ICs instead of inferring them from scratch. The idea of cascade connections is implemented by adding skip connections that combine the output of the base model hidden layer $f_{\theta_m}$ with the logits of $\text{IC}_{m-1}$ and pass it to $\text{IC}_m$. The prediction is realized by the softmax function applied to $g_{\phi_m}$ (the $m$-th IC network):

$$p_m = \text{softmax}(g_{\phi_m}(f_{\theta_m}(x), g_{\phi_{m-1}} \circ f_{\theta_{m-1}}(x))), \text{ for } m > 1, \tag{1}$$

where $g \circ f(x) = g(f(x))$ denotes the composition of functions. Formally, $p_m = p_m(x; \phi_m)$, where $\phi_m$ are trainable parameters of $\text{IC}_m$, but we drop these parameters in notation for brevity. $\text{IC}_1$ uses only the information coming from the layer $f_{\theta_1}$ which does not need to be the first hidden layer of $f_\theta$. Figure 1 shows the skip connections as red horizontal arrows.

Each $IC_m$ is trained in parallel (with respect to $\phi_m$) to optimize the prediction of all output classes using an appropriate loss function $\mathcal{L}(p_m)$, e.g. cross-entropy for classification. However, during the backward step it is crucial to stop the gradient of a loss function from passing to the previous classifier. Allowing the gradients of loss $\mathcal{L}(p_m)$ to affect $\phi_j$ for $j \in 1, .., m-1$ leads to a significant performance degradation of earlier layers due to increased focus on the features important for $IC_m$, as we show in Appendix C.3.

**Ensembling**   Ensembling in machine learning models reliably increases the performance of a model while improving robustness and uncertainty estimation [8, 23]. The main drawback of this approach is its wastefulness, as it requires to train multiple models and use them to process the same examples. However, in our setup we can adopt this idea to combine predictions which were already pre-computed in previous ICs, with near-zero additional computational cost.

To obtain a reliable zero-waste system, we build ensembles that combine outputs from groups of ICs to provide the final answer of the $m$-th classifier. Since the classifiers we are using vary significantly in predictive strength (later ICs achieve better performance than early ICs) and their predictions are correlated, the standard approach to deep model ensembling does not work in our case. Thus, we introduce weighted geometric mean with class balancing, which allows us to reliably find a combination of pre-computed responses that maximizes the expected result.

Let $p_1, p_2, \ldots, p_m$ be the outputs of $m$ consecutive IC predictions (after cascade connections stage) for a given $x$ (Figure 1). We define the probability of the $i$-th class in the $m$-th ensemble to be:

$$q_m^i(x) = \frac{1}{Z_m} b_m^i \prod_{j \leq m} \left( p_j^i(x) \right)^{w_m^j}, \tag{2}$$

where $b_m^i > 0$ and $w_m^j > 0$, for $j = 1, \ldots, m$, are trainable parameters, and $Z_m$ is a normalization factor, such that $\sum_i q_m^i(x) = 1$. Observe that $w_m^j$ can be interpreted as our prior belief in predictions of $IC_j$, i.e. large weight $w_m^j$ indicates less confidence in the predictions of $IC_j$. On the other hand, $b_m^i$ represents the prior of $i$-th class for $IC_m$. The $m$ indices in $w_m$ and $b_m$ are needed as the weights are trained independently for each subset $\{IC_j : j \leq m\}$. Although there are viable potential approaches to setting these parameters by hand, we verified that optimizing them directly by minimizing the cross-entropy loss on the training dataset works best.

Out of additive and geometric ensemble settings we found the latter to be preferable. In this formulation, a low class confidence of a single IC would significantly reduce the probability of that class in the whole ensemble. In consequence, in order for the confidence of the given class to be high, we require all ICs to be confident in that class. Thus, in geometric ensembling, an incorrect although confident IC answer has less chance of ending calculations prematurely. In the additive setting, the negative impact of a single confident but incorrect IC is much higher, as we show in Appendix C.2. Hence our choice of geometric ensembling.

Direct calculation of the product in (2) might lead to numerical instabilities whenever the probabilities are close to zero. To avoid this problem we note that

$$b_m^i \prod_{j \leq m} \left( p_j^i(x) \right)^{w_m^j} = b_m^i \exp \left( \sum_{j \leq m} w_m^j \ln p_j^i(x) \right),$$

and that log-probabilities $\ln p_j^i$ can be obtained by running the numerically stable log softmax function on the logits $g_{\phi_m}$ of the classifier.

Both cascade connections and ensembling have different impact on the model. Cascade connections primarily boost the accuracy of early ICs. Ensembling, on the other hand, improves primarily the performance of later ICs, which combine the information from many previous classifiers.

This is not surprising, given that the power of the ensemble increases with the number of members, provided they are at least weak in the sense of boosting theory [33]. As such, the two techniques introduced above are complementary, which we also show empirically via ablation studies in Appendix C. The whole training procedure is presented in Algorithm 1.

**Conditional inference**   Once a ZTW model is trained, the following question appears: how to use the constructed system at test time? More precisely, we need to dynamically find the shortest processing path for a given input example. For this purpose, we use one of the standard confidence

---

**Algorithm 1** Zero Time Waste

---

**Input:** pre-trained model $f_\theta$, cross-entropy loss function $\mathcal{L}$, training set $\mathcal{T}$.
**Initialize** $M$ shallow models $g_{\phi_m}$ at selected layers $f_{\theta_m}$.
**For** $m = 1, \ldots, M$ **do in parallel**     $\triangleright$ Cascade connection ICs
  Set $p_m$ according to (1).
  minimize $\mathbb{E}_{(x,y)\in\mathcal{T}} \left[ \mathcal{L}(p_m(x), y) \right]$ wrt. $\phi_m$ by gradient descent

**For** $m = 1, \ldots, M$ **do**     $\triangleright$ Geometric Ensembling
  Initialize $w_m, b_m$ and define $q_m(x)$ according to (2).
  minimize $\mathbb{E}_{(x,y)\in\mathcal{T}} \left[ \mathcal{L}(q_m(x), y) \right]$ wrt. $w_m, b_m$ by gradient descent

---

scores given by the probability of the most confident class. If the $m$-th classifier is confident enough about its prediction, i.e. if

$$\max_i q_m^i > \tau, \text{ for a fixed } \tau > 0, \tag{3}$$

where $i$ is the class index, then we terminate the computation and return the response given by this IC. If this condition is not satisfied, we continue processing $x$ and go to the next IC.

Threshold $\tau$ in (3) is a manually selected value, which controls the acceleration-performance trade-off of the model. A lower threshold leads to a significant speed-up at the cost of a possible drop in accuracy. Observe that for $\tau > 1$, we recover the original model $f_\theta$, since none of the ICs is confident enough to answer earlier. In practice, to select its appropriate value, we advise using a held-out set to evaluate a range of possible values of $\tau$.

## 4 Experiments

In this section we examine the performance of Zero Time Waste and analyze its impact on waste reduction in comparison to two recently proposed early-exit methods: (1) Shallow-Deep Networks (SDN) [19] and (2) Patience-Based Early Exit (PBEE) [47]. In contrast to SDN and PBEE, which train ICs independently, ZTW reuses information from past classifiers to improve the performance. SDN and ZTW use maximum class probability as the confidence estimator, while PBEE checks the number of classifiers in sequence that gave the same prediction. For example, for PBEE $\tau = 2$ means that if the answer of the current IC is the same as the answers of the 2 preceding ICs, we can return that answer, otherwise we continue the computation.

In our experiments, we measure how much computation we can save by re-using responses of ICs while keeping good performance, hence obeying the zero waste paradigm. To evaluate the efficiency of the model, we compute the average number of floating-point operations required to perform the forward pass for a single sample. We use it as a hardware-agnostic measure of inference cost and refer to it simply as the "inference time" in all subsequent references. For the evaluation in supervised learning, we use three datasets: CIFAR-10, CIFAR-100, and Tiny ImageNet, and four commonly used architectures: ResNet-56 [11], MobileNet [15], WideResNet [45], and VGG-16BN [37] as base networks. We check all combinations of methods, datasets, and architectures, giving $3 \cdot 3 \cdot 4 = 36$ models in total, and we additionally evaluate a single architecture on the ImageNet dataset to show that the approach is scalable. Additionally, we examine how Zero Time Waste performs at reducing waste in a reinforcement learning setting of Atari 2600 environments. To the best of our knowledge, we are the first to apply early exit methods to reinforcement learning.

Appendix A.1 describes the details about the network architecture, hyperparameters, and training process. Appendix B contains extended plots and tables, and results of an additional transfer learning experiment. In Appendix C we provide ablation studies, focusing in particular on analyzing how each of the proposed improvements affects the performance, and empirically justifying some of the design choices (e.g. geometric ensembles vs. additive ensembles). We provide the source code for our experiments at `https://github.com/gmum/Zero-Time-Waste`.

### 4.1 Time Savings in Supervised Learning

We check what percentage of computation of the base network can be saved by reusing the information from previous layers in a supervised learning setting. To do this, we evaluate how each method behaves at a particular fraction of the computational power (measured in floating point operations) of

Table 1: Results on four different architectures and three datasets: Cifar-10, Cifar-100 and Tiny ImageNet. Test accuracy (in percentages) for time budgets: 25%, 50%, 75%, 100% of the base network, and Max without any time limits. The first column shows the test accuracy of the base network. The results represent a mean of three runs and standard deviations are provided in Appendix B. We bold results within two standard deviations of the best model.

| | | **ResNet-56** | | | | | | **MobileNet** | | | |
|---|---|---|---|---|---|---|---|---|---|---|---|
| Data | Algo | 25% | 50% | 75% | 100% | Max | Data | Algo | 25% | 50% | 75% | 100% | Max |
| **C10** (92.0) | SDN | 77.7 | 87.3 | 91.1 | **92.0** | **92.1** | **C10** (90.6) | SDN | **86.1** | **90.5** | 90.8 | 90.7 | 90.9 |
| | PBEE | 69.8 | 81.8 | 87.5 | 91.0 | **92.1** | | PBEE | 76.3 | 85.9 | 89.7 | 90.9 | 91.1 |
| | ZTW | **80.3** | **88.7** | **91.5** | **92.1** | **92.1** | | ZTW | **86.7** | **90.9** | **91.4** | **91.4** | **91.5** |
| **C100** (68.4) | SDN | 47.1 | 57.2 | 64.7 | 69.0 | 69.7 | **C100** (65.1) | SDN | **54.3** | 63.5 | 66.8 | 67.8 | 67.9 |
| | PBEE | 45.2 | 53.5 | 60.1 | 67.0 | 69.0 | | PBEE | 47.1 | 61.6 | 61.6 | 67.0 | 68.0 |
| | ZTW | **51.3** | **62.1** | **68.4** | **70.7** | **70.9** | | ZTW | **54.5** | **65.2** | **68.4** | **69.0** | **69.1** |
| **T-IM** (53.9) | SDN | 31.2 | 41.2 | 49.9 | 54.5 | 54.7 | **T-IM** (59.3) | SDN | **35.6** | 47.1 | 55.3 | 58.9 | 59.7 |
| | PBEE | 29.0 | 37.6 | 48.2 | 53.4 | 54.3 | | PBEE | 26.7 | 38.4 | 50.3 | 55.6 | 59.7 |
| | ZTW | **35.2** | **46.2** | **53.7** | **56.3** | **56.4** | | ZTW | **37.3** | **49.5** | **56.7** | **59.7** | **60.2** |

| | | **WideResNet** | | | | | | **VGG** | | | |
|---|---|---|---|---|---|---|---|---|---|---|---|
| Data | Algo | 25% | 50% | 75% | 100% | Max | Data | Algo | 25% | 50% | 75% | 100% | Max |
| **C10** (94.4) | SDN | 83.8 | 91.7 | 94.1 | 94.4 | 94.4 | **C10** (93.0) | SDN | 86.0 | 92.1 | **93.0** | **93.0** | **93.0** |
| | PBEE | 78.0 | 84.0 | 90.3 | 93.8 | 94.4 | | PBEE | 75.0 | 86.0 | 91.0 | 92.9 | **93.1** |
| | ZTW | **86.7** | **92.9** | **94.5** | **94.7** | **94.7** | | ZTW | **87.1** | **92.5** | **93.2** | **93.2** | **93.2** |
| **C100** (75.1) | SDN | 55.9 | 65.1 | 71.6 | 75.0 | 75.4 | **C100** (70.4) | SDN | 58.5 | 67.2 | 70.6 | 71.4 | 71.5 |
| | PBEE | 46.7 | 57.2 | 66.0 | 73.2 | 75.4 | | PBEE | 51.2 | 65.3 | 65.3 | 70.9 | 72.0 |
| | ZTW | **59.5** | **69.1** | **74.5** | **76.2** | **76.4** | | ZTW | **60.2** | **69.3** | **72.6** | **73.5** | **73.6** |
| **T-IM** (59.6) | SDN | 36.8 | 46.0 | 54.6 | 59.4 | 59.7 | **T-IM** (59.0) | SDN | 40.0 | 50.5 | 57.4 | **59.6** | 59.7 |
| | PBEE | 29.9 | 37.8 | 52.7 | 58.5 | 59.7 | | PBEE | 31.0 | 45.2 | 55.2 | **60.1** | **60.2** |
| | ZTW | **40.0** | **50.1** | **57.5** | **60.2** | **60.3** | | ZTW | **41.4** | **52.3** | **59.3** | **60.1** | **60.5** |

the base network. We select the highest threshold $\tau$ such that the average inference time is smaller than, for example, 25% of the original time. Then we calculate accuracy for that threshold. Table 1 contains summary of this analysis, averaged over three seeds, with further details (plots for all thresholds, standard deviations) shown in Appendix B.1.

Looking at the results, we highlight the fact that methods which do not reuse information between ICs do not always achieve the goal of reducing computational waste. For example, SDN and PBEE cannot maintain the accuracy of the base network for MobileNet on Tiny ImageNet when using the same computational power, scoring respectively 0.4 and 3.7 percentage points lower than the baseline. Adding ICs to the network and then discarding their predictions when they are not confident enough to return the final answer introduces computational overhead without any gains. By reusing the information from previous ICs ZTW overcomes this issue and maintains the accuracy of the base network for all considered settings. In particular cases, such as ResNet-56 on Tiny ImageNet or MobileNet on Cifar-100, Zero Time Waste even significantly outperforms the core network.

Similar observation can be made for other inference time limits as well. ZTW consistently maintains high accuracy using less computational resources than the other approaches, for all combinations of datasets and architectures. Although PBEE reuses information from previous layers to decide whether to stop computation or not, this is not sufficient to reduce the waste in the network. While PBEE outperforms SDN when given higher inference time limits, it often fails for smaller limits (25%, 50%). We hypothesize that this is result of the fact that PBEE has smaller flexibility with respect to $\tau$. While for SDN and ZTW values of $\tau$ are continuous, for PBEE they represent a discrete number of ICs that must sequentially agree before returning an answer.

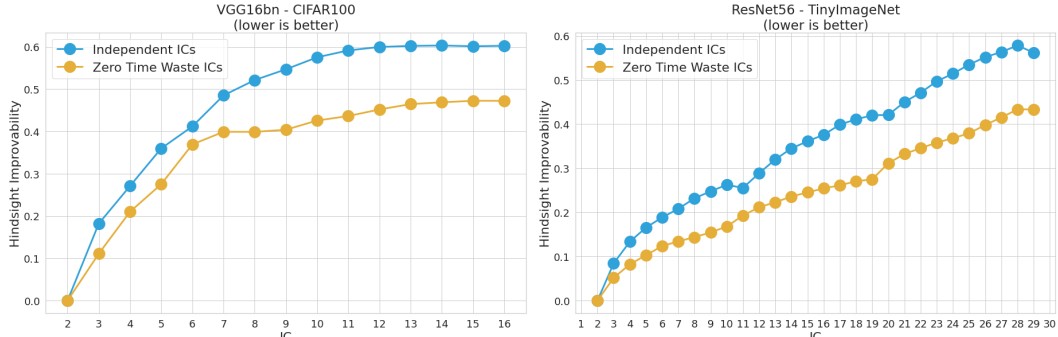

Figure 2: Hindsight Improvability. For each IC (horizontal axis) we look at examples it misclassified and we check how many of them were classified correctly by any of the previous ICs. The lower the number, the better the IC is at reusing previous information.

Finally, we check whether our observations scale up to larger datasets by running experiments on ImageNet using a pre-trained ResNet-50 from the torchvision package[3]. The results presented in Table 2 show that Zero Time Waste is able to gain significant improvements over the two tested baselines even in this more challenging setting. Additional details of this experiments are presented in Appendix B.2.

Given the performance of ZTW, the results show that paying attention to the minimization of computational waste leads to tangible, practical improvements of the inference time of the network. Therefore, we devote next section to explaining where the empirical gains come from and how to measure information loss in the models.

Table 2: ImageNet results (test accuracy in percentage points) show that zero-waste approach scales up to larger datasets.

| Algo | 25% | 50% | 75% | 100% |
|------|------|------|------|------|
| SDN  | 33.8 | 53.8 | 69.7 | 75.8 |
| PBEE | 28.3 | 28.3 | 62.9 | 73.3 |
| ZTW  | **34.9** | **54.9** | **70.6** | **76.3** |

## 4.2 Information Loss in Early Exit Models

Since ICs in a given model are heavily correlated, it is not immediately obvious why reusing past predictions should improve performance. Later ICs operate on high-level features for which class separation is much easier than for early ICs, and hence get better accuracy. Thus, we ask a question — is there something that early ICs know that the later ICs do not?

For that purpose, we introduce a metric to evaluate how much a given IC could improve performance by reusing information from all previous ICs. We measure it by checking how many examples incorrectly classified by $IC_m$ were classified correctly by any of the previous ICs. An IC which reuses predictions from the past perfectly would achieve a low score on this metric since it would remember all the correct answers of the previous ICs. On the other hand, an IC in a model which trains each classifier independently would have a higher score on this metric, since it does not use past information at all. We call this metric Hindsight Improvability (HI) since it measures how many mistakes we would be able to avoid if we used information from the past efficiently.

Let $\mathcal{C}_m$ denote the set of examples correctly classified by $IC_m$, with its complement $\overline{\mathcal{C}}_m$ being the set of examples classified incorrectly. To measure the Hindsight Improvability of $IC_m$ we calculate:

$$\text{HI}_m = \frac{|\overline{\mathcal{C}}_m \cap (\bigcup_{n<m} \mathcal{C}_n)|}{|\overline{\mathcal{C}}_m|}$$

Figure 2 compares the values of HI for a method with independent ICs (SDN in this case) and ZTW which explicitly recycles computations. In the case of VGG16 trained with independent ICs, over $60\%$ of the mistakes could be avoided if we properly used information from the past, which

---

[3]https://pytorch.org/vision/stable/index.html

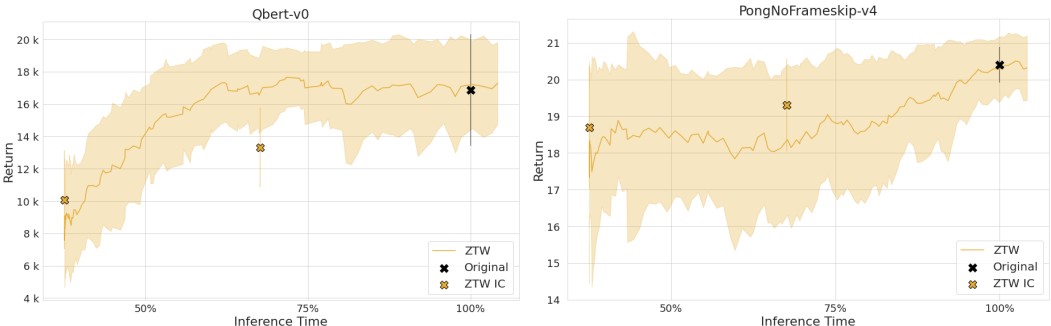

Figure 3: Inference time vs. average return of the ZTW policy in an RL setting on Q*bert and Pong Atari 2600 environments. The plot was generated by using different values of the confidence threshold $\tau$ hyperparameter. Since the RL environments are stochastic, we plot the return with a standard deviation calculated on 10 runs. ZTW saves a significant amount of computation while preserving the original performance, showcasing that waste can be minimized also in the reinforcement learning domain.

would translate to improvement from $71.5\%$ to $82.9\%$ accuracy. Similarily, for ResNet-56 trained on TinyImageNet, the number of errors could be cut by around $57\%$.

ZTW consistently outperforms the baseline, with the largest differences visible at the later ICs, which can in principle gain the most from reusing previous predictions. Thus, Zero Time Waste is able to efficiently recycle information from the past. At the same time, there is still a room for significant improvements, which shows that future zero waste approaches could offer additional enhancements.

## 4.3   Time Savings in Reinforcement Learning

Although supervised learning is an important testbed for deep learning, it does not properly reflect the challenges encountered in the real world. In order to examine the impact of waste-minimization methods in a setting that reflects the sequential nature of interacting with the world, we evaluate it in a Reinforcement Learning (RL) setting. In particular, we use the environments from the suite of Atari 2600 games [28].

Similarly as in the supervised setting, we start with a pre-trained network, which in this case represents a policy trained with the Proximal Policy Optimization (PPO) algorithm [36]. We attach the ICs to the network and train it by distilling the knowledge from the core network to the ICs. We use a behavioral cloning approach, where the states are sampled from the policy defined by the ICs and the labels are provided by the expert model. Since actions in Atari 2600 are discrete, we can then use the same confidence threshold-based approach to early exit inference as in the case of classification. More details about the training process are provided in the Appendix A.2.

In order to investigate the relationship between computation waste reduction and performance, we evaluate Zero Time Waste for different values of confidence threshold $\tau$. By setting a very high $\tau$ value, we retrieve the performance of the original model (none of the ICs respond) and by slowly decreasing its value we can reduce the computational cost (ICs begin to return answers earlier). In Figure 3 we check values of $\tau$ in the interval $[0.1, 1.0]$ to show how ZTW is able to control the acceleration-performance balance for Q*Bert and Pong, two popular Atari 2600 environments. By setting lower $\tau$ thresholds for Q*Bert we can save around $45\%$ of computations without score degradation. Similarly, for Pong we can get $60\%$ reduction with minor impact on performance (note that average human score is 9.3 points). This shows that even the small four-layered convolutional architecture commonly used for Atari [28] introduces a noticeable waste of computation which can be mitigated within a zero-waste paradigm. We highlight this fact as the field of reinforcement learning has largely focused on efficiency in terms of number of samples and training time, while paying less attention to the issue of efficient inference.

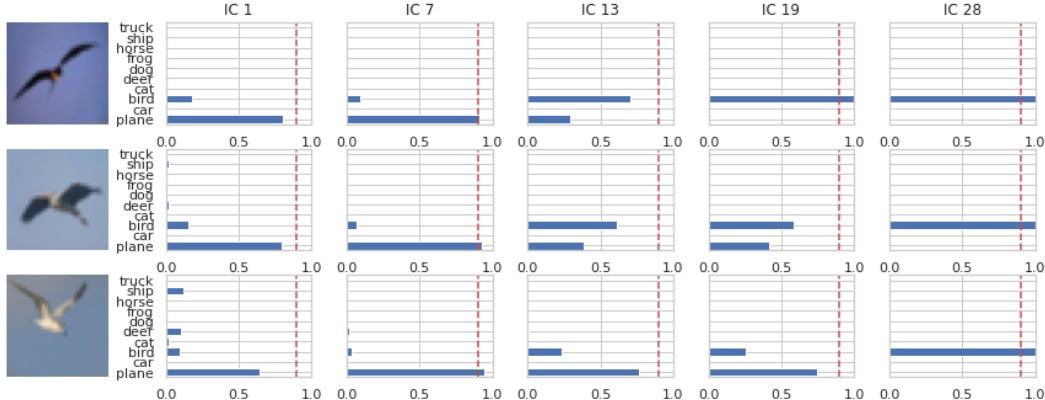

Figure 4: Examples of bird images which were incorrectly classified as airplanes by ZTW. The early ICs are misled by the low-level features (blue sky, sharp edges, grayscale silhouette) and return a prediction before the later ICs can detect more subtle high-level features.

### 4.4 Impact & Limitations

Our framework is the cornerstone of an environmental-aware computation where information recycling within a model is cautiously studied to avoid wasting resources. The focus on computational efficiency, however, introduces a natural trade-off between model accuracy and its computational cost. Although in most cases we can carefully adjust the appropriate method hyperparameters to avoid a significant accuracy drop, some testing samples remain surprisingly challenging for ZTW, which indicates a need for further investigation of the accuracy vs. computation cost trade-off offered by our method.

Figure 4 contains examples of images for which low-level features in a given image consistently point at a wrong class, while high-level features would allow us to deduce the correct class. Images of birds which contain sharp lines and grayscale silhouettes are interpreted as airplanes by early ICs which operate on low-level features. If the confidence of these classifiers gets high enough, the answer might be returned before later classifiers can correct this decision. We highlight the problem of dealing with examples which are seemingly easy but turn out difficult as an important future direction for conditional computation methods.

## 5 Conclusion

In this work, we show that discarding predictions of the previous ICs in early exit models leads to waste of computation resources and a significant loss of information. This result is supported by the introduced Hindsight Improvability metric, as well as empirical result for reducing computations in existing networks. The proposed Zero Time Waste method attempts to solve these issues by incorporating outputs from the past heads by using cascade connections and geometric ensembling. We show that ZTW outperforms other approaches on multiple standard datasets and architectures for supervised learning, as well as in Atari 2600 reinforcement learning suite. At the same time we postulate that focusing on reducing the computational waste in a safe and stable way is an important direction for future research in deep learning.

## Acknowledgments & Funding Disclosure

This research was funded by Foundation for Polish Science (grant no POIR.04.04.00-00-14DE/18-00 carried out within the Team-Net program co-financed by the European Union under the European Regional Development Fund) and National Science Centre, Poland (grant no 2018/31/B/ST6/00993 and grant no 2020/39/B/ST6/01511). The authors have applied a CC BY license to any Author Accepted Manuscript (AAM) version arising from this submission, in accordance with the grants' open access conditions.

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
