# A    Training Details

All experiments were performed using a single Tesla V100 GPU.

## A.1    Supervised Learning

We setup the core networks in our CIFAR-10, CIFAR-100, and Tiny ImageNet experiments follow-ing [19] for fair comparison. We use these trained networks and treat them as pre-trained models, i.e. we consider the „IC-only" setup, where we do not change the base network.

For CIFAR-10 and CIFAR-100 we train ICs for 50 epochs using the Adam optimizer with learning rate set to $0.001$, but lowered by a factor of 10 after 15 epochs. When training on Tiny ImageNet, the learning rate is additionally lowered again by the same factor after epoch 40. On ImageNet (on the pretrained ResNet-50 from the *torchvision* package), the ICs are trained for $40$ epochs, with the initial learning rate of $0.00001$ being reduced by a factor of 10 in epochs 20 and 30. To train the ensembling part of our method, we run SGD on the training dataset for 500 epochs. Since both the dataset and the model are very small, we use a high number of epochs to ensure convergence.

**Architecture and Placement of ICs**    Most common computer vision architectures, including the ones we use, are divided into blocks (e.g. residual blocks in ResNet). Because some blocks change the dimensionality of the features, we take the natural choice of attaching an IC after each block, which also considerably simplifies the implementation of our method for any future architectures. Note that the resulting uniform distribution of ICs along the base network is not necessarily optimal [32]. However, we focus on this setup for the sake of a fair comparison with SDN and PBEE and consider the exploration of the best placement of ICs as outside the scope of this work.

Each IC consists of a single convolutional layer, a pooling layer, and a fully-connected layer, which outputs the class logits. The convolutional layer has a kernel size of 3 with the number of output filters equal to the number of input channels. When applying cascade connections in Zero Time Waste, we use the outputs of the previous IC as an additional input to the linear classification layer of the current IC, as shown earlier in Figure 1. Because Tiny ImageNet has a larger input image size than CIFAR datasets, we use convolutions with stride 2 instead of 1 to reduce the number of operations of each IC.

For the pooling layer we reuse the SDN pooling proposed by [19], which is defined as:

$$\text{sdn\_pool}(x) = \gamma \cdot \text{avg\_pool}(x) + (1 - \gamma) \cdot \text{max\_pool}(x),$$

where $\gamma$ is a learnable scalar parameter. It reduces the size of convolutional maps to $4 \times 4$.

We keep the architecture and IC placement fixed between experiments, but with small exceptions for Tiny ImageNet and ImageNet. For Tiny ImageNet, we use convolutional layers with stride set to 2 if all dimensions of the input are larger than 8. We do the same for ImageNet, but we additionally reduce the number of output channels of that convolution by a factor of $4$ and we place ICs only every third ResNet block. Finally, we apply Layer Normalization to the output of the preceding IC before using it in the final linear layer.

## A.2    Reinforcement Learning

We set the Atari environments as follows. Every fourth frame (frame skipping) and the one immedi-ately before it are max-pooled. The resulting frame is then rescaled to size 84x84 and converted into grayscale. At every step the agent has a $0.1$ probability of taking the previous action irrespective of the policy probabilities (sticky actions). This is added to introduce stochasticity into the environment to avoid cases when the policy converges to a simple strategy that results in the same actions taken in every run. Furthermore, the environment termination flag is set when a life is lost. Finally, the signum function of the reward is taken (reward clipping). The above setup is fairly common and we base our code on the popular Stable Baselines repository [30].

Using that environment setup we use the PPO algorithm to train the policy, and then extract the base network by discarding the value network. We use the following PPO hyperparameters: learning rate $2.5 \cdot 10^{-4}$, 128 steps to run for each environment per update, batch size 256, 4 epochs of surrogate loss optimization, clip range ($\epsilon$) 0.1, entropy coefficient 0.01, value function coefficient 0.5, discount factor 0.99, 0.95 as the trade-off of bias vs variance factor for Generalized Advantage Estimator [35],

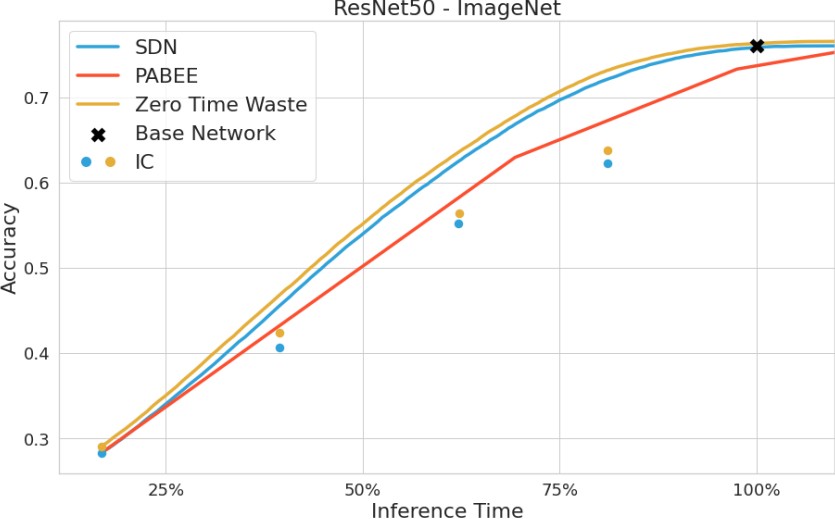

Figure 5: Inference time vs. accuracy for ResNet-50 trained on ImageNet. Base network achieves 76.0% accuracy, and given the same inference time constraint SDN obtains 75.8%, PBEE 73.3%, and ZTW 76.3%.

and the maximum value for the gradient clipping $0.5$. The policy is trained for $10^7$ environment time steps in total.

We use the standard 'NatureCNN' [28] architecture with three convolutional layers and a single fully connected layer. We attach two ICs after the first and the second layer. Similarly as in the supervised setting, each IC has a single convolutional layer, an SDN pooling layer and a fully connected layer. The convolutional layer has stride set to $4$ and preserves the number of channels.

To train the ICs, the early-exit policy interacts with the environment. In each step, an IC is chosen uniformly, and the action chosen by that IC is taken. However, the $(o, a_p)$ tuple is actually saved to the replay buffer, with $o$ and $a_p$ being the observation and the action of the original policy, respectively. After $128$ concurrent steps on $8$ environments that buffer is used to train the ICs with behavioral cloning. That is, Kullback–Leibler divergence between the PPO policy actions and the IC actions is used as the cost function. This is done for $5$ epochs with batch size set to $64$ and $128$ for cascading stage and geometric ensembling stage, respectively. The entire process is repeated until $10^6$ or more steps in total are taken.

# B   Additional results

This section contains experimental results which were omitted in the main part of the paper due to page limitations.

## B.1   Supervised Learning

For brevity, in the main part of the paper we have only shown a table summarizing the results of acceleration on multiple architectures and dataset. Here, we provide a fuller representation of these results. Figures 10, 11 and 12 (at the end of the Appendix) show results of the tested methods on CIFAR-10, CIFAR-100 and Tiny ImageNet, respectively. Each figure contains plots for the four considered architectures: ResNet-56, MobileNet, WideResNet and VGG16. Plots show that ZTW outperforms SDN and PBEE in almost all settings, which is consistent with the results summarized earlier. Additionally, in Table 3 we provide summary of the results with standard deviations. Figures 13, 14, 15 show values of Hindsight Improvability for CIFAR-10, CIFAR-100 and Tiny ImageNet, respectively.

Table 3: Results on four different architectures and three datasets: Cifar-10, Cifar-100 and Tiny ImageNet. Test accuracy (in percentages) obtained using the time budget: 25%, 50%, 75%, 100% of the base network and Max without any limits. The first column shows the test accuracy of the base network.

**ResNet-56**

| Data | Algo | 25% | 50% | 75% | 100% | Max |
|---|---|---|---|---|---|---|
| **CIFAR-10** (92.0 ± 0.2) | SDN | 77.7 ± 1.0 | 87.3 ± 0.5 | 91.1 ± 0.2 | **92.0 ± 0.1** | 92.1 ± 0.2 |
| | PBEE | 69.8 ± 1.3 | 81.8 ± 0.3 | 87.5 ± 0.1 | 91.0 ± 0.3 | **92.1 ± 0.3** |
| | ZTW | **80.3 ± 1.0** | **88.7 ± 0.4** | **91.5 ± 0.2** | 92.1 ± 0.3 | 92.1 ± 0.3 |
| **CIFAR-100** (68.4 ± 0.2) | SDN | 47.1 ± 0.2 | 57.2 ± 0.4 | 64.7 ± 0.6 | 69.0 ± 0.2 | 69.7 ± 0.2 |
| | PBEE | 45.2 ± 0.5 | 53.5 ± 0.5 | 60.1 ± 0.5 | 67.0 ± 0.2 | 69.0 ± 0.2 |
| | ZTW | **51.3 ± 0.4** | **62.1 ± 0.3** | **68.4 ± 0.4** | **70.7 ± 0.1** | **70.9 ± 0.1** |
| **Tiny ImageNet** (53.9 ± 0.3) | SDN | 31.2 ± 0.2 | 41.2 ± 0.3 | 49.9 ± 0.4 | 54.5 ± 0.5 | 54.7 ± 0.4 |
| | PBEE | 29.0 ± 0.6 | 37.6 ± 0.3 | 48.2 ± 0.4 | 53.4 ± 0.6 | 54.3 ± 0.4 |
| | ZTW | **35.2 ± 0.7** | **46.2 ± 0.4** | **53.7 ± 0.3** | **56.3 ± 0.3** | **56.4 ± 0.3** |

**MobileNet**

| Data | Algo | 25% | 50% | 75% | 100% | Max |
|---|---|---|---|---|---|---|
| **CIFAR-10** (90.6 ± 0.2) | SDN | **86.1 ± 0.5** | **90.5 ± 0.2** | 90.8 ± 0.1 | 90.7 ± 0.2 | 90.9 ± 0.1 |
| | PBEE | 76.3 ± 0.9 | 85.9 ± 0.3 | 89.7 ± 0.3 | 90.9 ± 0.2 | 91.1 ± 0.1 |
| | ZTW | **86.7 ± 0.7** | **90.9 ± 0.3** | **91.4 ± 0.2** | **91.4 ± 0.1** | **91.5 ± 0.1** |
| **CIFAR-100** (65.1 ± 0.3) | SDN | **54.3 ± 1.4** | 63.5 ± 0.8 | 66.8 ± 0.4 | 67.8 ± 0.1 | 67.9 ± 0.1 |
| | PBEE | 47.1 ± 2.7 | 61.6 ± 0.7 | 61.6 ± 0.7 | 67.0 ± 0.3 | 68.0 ± 0.3 |
| | ZTW | **54.5 ± 1.1** | **65.2 ± 0.5** | **68.4 ± 0.3** | **69.0 ± 0.1** | **69.1 ± 0.1** |
| **Tiny ImageNet** (59.3 ± 0.1) | SDN | 35.6 ± 1.3 | 47.1 ± 0.6 | 55.3 ± 0.3 | 58.9 ± 0.2 | 59.7 ± 0.1 |
| | PBEE | 26.7 ± 1.5 | 38.4 ± 2.0 | 50.3 ± 0.8 | 55.6 ± 0.3 | 59.7 ± 0.0 |
| | ZTW | **37.3 ± 2.8** | **49.5 ± 1.9** | **56.7 ± 0.6** | **59.7 ± 0.4** | **60.2 ± 0.1** |

**WideResNet**

| Data | Algo | 25% | 50% | 75% | 100% | Max |
|---|---|---|---|---|---|---|
| **CIFAR-10** (94.4 ± 0.1) | SDN | 83.8 ± 1.3 | 91.7 ± 0.5 | 94.1 ± 0.2 | 94.4 ± 0.1 | 94.4 ± 0.2 |
| | PBEE | 78.0 ± 1.9 | 84.0 ± 1.4 | 90.3 ± 0.5 | 93.8 ± 0.1 | 94.4 ± 0.1 |
| | ZTW | **86.7 ± 0.7** | **92.9 ± 0.3** | **94.5 ± 0.1** | **94.7 ± 0.1** | **94.7 ± 0.1** |
| **CIFAR-100** (75.1 ± 0.1) | SDN | 55.9 ± 1.5 | 65.1 ± 0.9 | 71.6 ± 0.4 | 75.0 ± 0.1 | 75.4 ± 0.1 |
| | PBEE | 46.7 ± 2.0 | 57.2 ± 1.3 | 66.0 ± 0.6 | 73.2 ± 0.2 | 75.4 ± 0.2 |
| | ZTW | **59.5 ± 0.6** | **69.1 ± 0.9** | **74.5 ± 0.6** | **76.2 ± 0.3** | **76.4 ± 0.2** |
| **Tiny ImageNet** (59.6 ± 0.6) | SDN | 36.8 ± 0.1 | 46.0 ± 1.0 | 54.6 ± 0.7 | 59.4 ± 0.8 | 59.7 ± 0.7 |
| | PBEE | 29.9 ± 0.9 | 37.8 ± 0.6 | 52.7 ± 0.6 | 58.5 ± 0.9 | 59.7 ± 0.7 |
| | ZTW | **40.0 ± 0.3** | **50.1 ± 0.2** | **57.5 ± 0.4** | **60.2 ± 0.1** | **60.3 ± 0.2** |

**VGG**

| Data | Algo | 25% | 50% | 75% | 100% | Max |
|---|---|---|---|---|---|---|
| **CIFAR-10** (93.0 ± 0.0) | SDN | 86.0 ± 0.3 | 92.1 ± 0.1 | **93.0 ± 0.0** | **93.0 ± 0.0** | **93.0 ± 0.0** |
| | PBEE | 75.0 ± 0.2 | 86.0 ± 0.2 | 91.0 ± 0.3 | 92.9 ± 0.2 | 93.1 ± 0.1 |
| | ZTW | **87.1 ± 0.1** | **92.5 ± 0.1** | **93.2 ± 0.2** | **93.2 ± 0.2** | **93.2 ± 0.2** |
| **CIFAR-100** (70.4 ± 0.3) | SDN | 58.5 ± 0.4 | 67.2 ± 0.1 | 70.6 ± 0.3 | 71.4 ± 0.2 | 71.5 ± 0.4 |
| | PBEE | 51.2 ± 0.2 | 65.3 ± 0.3 | 65.3 ± 0.3 | 70.9 ± 0.5 | 72.0 ± 0.4 |
| | ZTW | **60.2 ± 0.2** | **69.3 ± 0.4** | **72.6 ± 0.1** | **73.5 ± 0.3** | **73.6 ± 0.4** |
| **Tiny ImageNet** (59.0 ± 0.2) | SDN | 40.0 ± 1.0 | 50.5 ± 0.2 | 57.4 ± 0.5 | **59.6 ± 0.3** | 59.7 ± 0.3 |
| | PBEE | 31.0 ± 1.6 | 45.2 ± 0.6 | 55.2 ± 0.3 | **60.1 ± 0.5** | **60.2 ± 0.5** |
| | ZTW | **41.4 ± 0.5** | **52.3 ± 0.4** | **59.3 ± 0.4** | 60.1 ± 0.5 | **60.5 ± 0.4** |

## B.2 Results of ImageNet experiments

In order to show that the proposed method scales up well to the ImageNet dataset, we use our method on a pre-trained model provided by the torchvision package[4]. The obtained model allows for

---

[4] https://pytorch.org/vision/stable/index.html

significant speed-ups on ImageNet while maintaining the same accuracy for the original inference time limit. The results presented in Figure 5 show that ZTW again outperforms the rest of the methods, with SDN maintaining reasonable, although lower, performance and PBEE generally failing. We want to highlight the fact that the architecture of ICs used here is very simple and nowhere as intensely investigated as the architecture of ResNet or other common deep learning models. Adjusting the ICs for this problem could thus improve the results significantly, although we consider this outside the scope of this work.

### B.3 Results of Transfer Learning experiments

We investigate whether early exit methods work in a transfer learning setting. We use ResNet-50 from the torchvision package pre-trained on ImageNet similarly as in the previous experiment. To obtain a baseline standard classifier, we remove the final linear layer of the pretrained classifier and train a new linear layer with the number of outputs corresponding to the number of classes in the target dataset. Only then we proceed to train the ICs.

Table 4: Results on the OCT2017 dataset when using an ImageNet pretrained core network. Test accuracy (in percentages) obtained using the time budget: 25%, 50%, 75%, 100% of the base network and Max without any limits.

| | **ResNet-50** (94.6) | | | | |
|---|---|---|---|---|---|
| Algo | 25% | 50% | 75% | 100% | Max |
| SDN | 81.5 | 93.8 | 94.6 | 94.6 | 94.6 |
| PBEE | 56.5 | 90.3 | 90.3 | 94.5 | 95.2 |
| ZTW | 89.4 | 98.0 | 98.4 | 98.5 | 98.5 |

We use the OCT-2017 medical dataset [20] as the target dataset. The training dataset consists of $83484$ high-resolution retinal optical coherence tomography images categorized into four classes, with one class meaning healthy sample, and three diseases. Table 4 shows that ZTW outperforms other methods by a significant margin, and manages to cut down the time required to obtain the accuracy of the baseline by over 75%. This suggests that leveraging the power of previous ICs is especially useful when the features are not perfectly adjusted to the problem at hand, i.e. were trained for ImageNet classification and used for pathology classification data from a completely different domain. We aim to explore the transfer learning setting in future work.

### B.4 Results of Reinforcement Learning experiments

In Figure 6 we show the results for all eight Reinforcement Learning environments that we ran our experiments on. Degree of time savings depends heavily on the environment. For some of the environments, such as AirRaid and Pong, the ICs obtain a similar return to that of the original policy. Because of that the resulting plot is almost flat, allowing for significant inference time reduction without any performance drop. Other environments, such as Seaquest, Phoenix and Riverraid, allow to gradually trade-off performance for inference time just as in the supervised setting.

## C   Ablation Studies

In this section, we present results of experiments which explain our design decisions. In particular, we focus here on four issues: (1) what is the individual impact of cascade connections and geometric ensembling, (2) how performance of additive and geometric ensembles compares in our setting, (3) how stopping the gradient in cascade connections impacts learning dynamics, and (4) how the number of classes in the training dataset impacts the results.

### C.1   Impact of cascading and ensembling

An important question is whether we need both components in the proposed model (cascade connections and ensembling), and what role do they play in the final performance of our model. Figure 7 shows the results of independently applied cascade connections and geometric ensembling on a ResNet-56 and VGG-16 trained on CIFAR-100. We observe that depending on the threshold $\tau$ and the architecture, one of these techniques may be more important than the other. However, combining

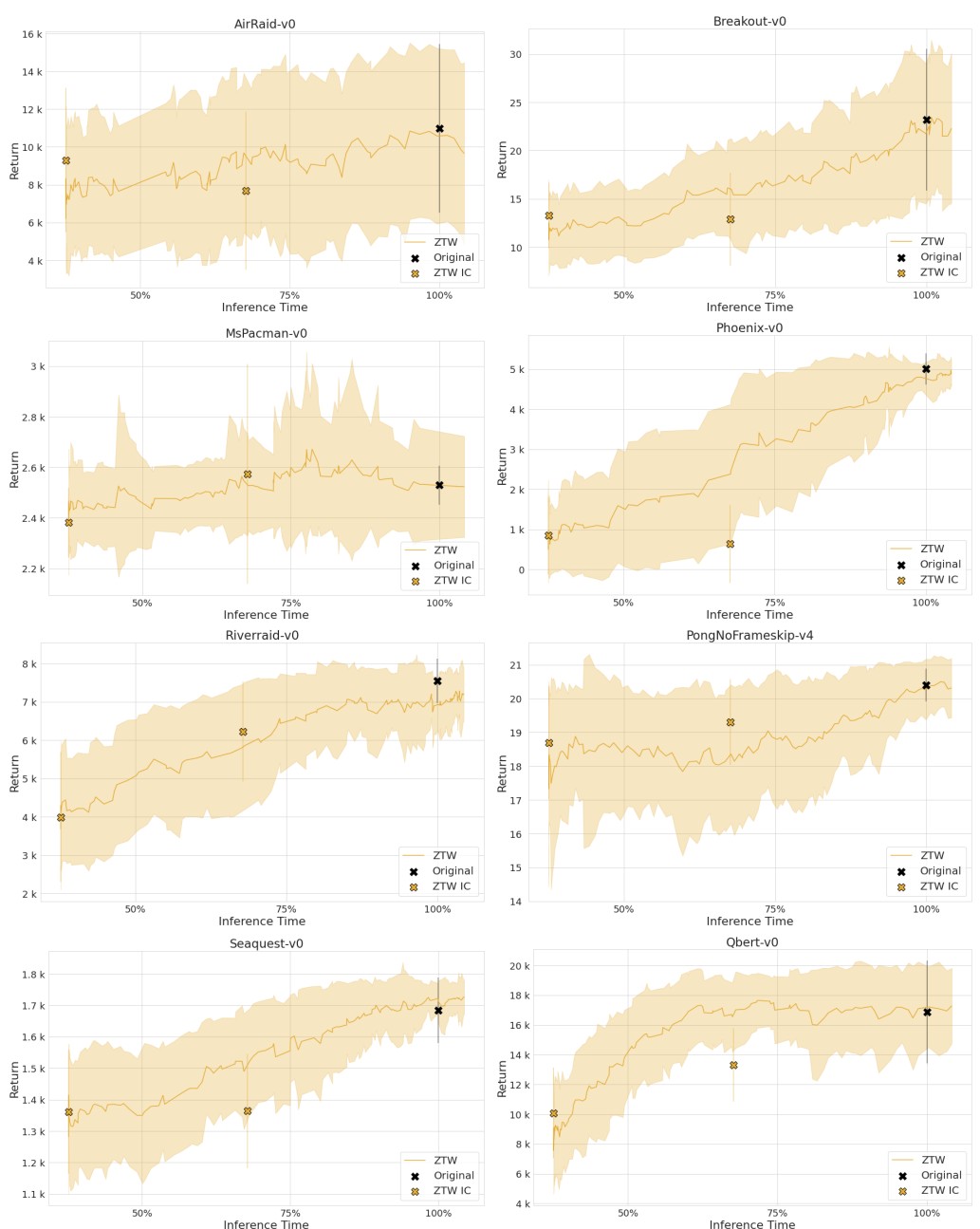

Figure 6: Mean and standard deviation of returns for multiple confidence thresholds on various Atari 2600 environments. Some environments allow significant computational savings with a negligible or no impact on performance.

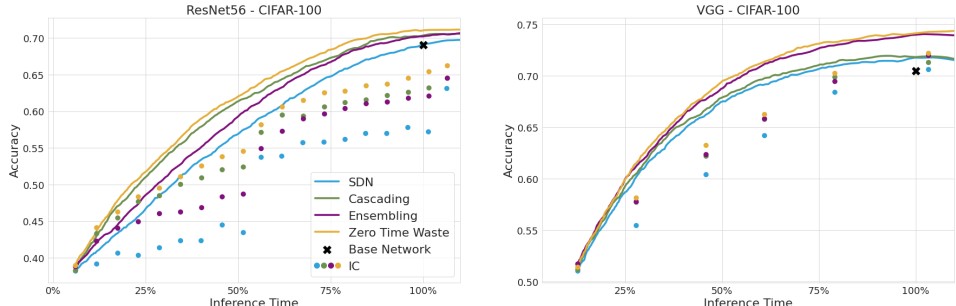

Figure 7: Ablation studies exhibiting the importance of both techniques proposed in the paper. Although both cascade connections and geometric ensembling seem to help, the exact effect depends on the architecture and chosen threshold $\tau$. For ResNet56 cascade connections seem to be much more helpful than ensembling, while for VGG16 the opposite is true. As such, both are required to consistently improve results.

these methods consistently improves the performance each of them achieved independently. Thus we argue that both cascade connections and geometric ensembling are required in Zero Time Waste and using only one of them will lead to significant performance deterioration.

## C.2    Geometric vs Additive Ensembles

In this work we proposed geometric ensembles for combining predictions from multiple ICs. Here, we show how this approach performs in comparison to additive ensemble of the form:

$$q^i_m(x) = \frac{1}{Z_m} \sum_{j \leq m} w^j_m p^i_j(x) + b^i_m, \tag{4}$$

where $b^i_m > 0$ and $w^j_m > 0$, for $j = 1, \ldots, m$, are trainable parameters, and $Z_m$ is a normalization value, such that $\sum_i q^i_m(x) = 1$. That is, we use the same approach as in geometric ensembles, but we substitute the product for a sum and change the weighting scheme.

The empirical comparison between an additive ensemble and a geometric ensemble on ResNet-56 is presented in Figure 8. The results show that the geometric ensemble consistently outperforms the additive ensemble, although the magnitude of improvement varies across datasets. While the difference on CIFAR-10 is negligible, it becomes evident on Tiny ImageNet, especially with the later layers. The results suggest that geometric ensembling is more helpful on more complex datasets with a larger number of classes.

## C.3    Stop gradients in cascade connections

As mentioned in Section 3 of the main paper, we decide to stop gradient from flowing through the cascade connections. We motivate this decision by noticing that the gradients of later layers might destroy the predictive power of the earlier layers. In order to test this hypothesis empirically, we run our experiments on ResNet-56, with and without gradient stopping. As shown in Figure 9, the accuracy of the early ICs is lower when not using gradient stopping. Performance of later ICs may vary, as not using stopping gradient allows greater expressivity for later ICs. Since the second component of our method, ensembling, is able to reuse information from the early ICs we find it beneficial to use gradient stopping in the final model. This is especially evident on Tiny ImageNet, where on later ICs cascade connections perform better without gradient stopping, but ZTW is able to reuse ICs trained with gradient stopping more effectively.

We provide a more in-depth observation of the reason why the gradient of later ICs might have a detrimental effect on the performance of early ICs. Observe that in the setting without the detach the parameters of the first IC will be updated using $\sum_k g_k$, where $g_k$ is the gradient of the loss of the $k$-th IC wrt. parameters of the first IC. Experimental investigation showed that the cosine similarity of $\sum_k g_k$ and $g_1$ is approximately $0.5$ at the beginning of the training, which means that these gradients point in different directions. Since the gradient $g_1$ represents the best direction for

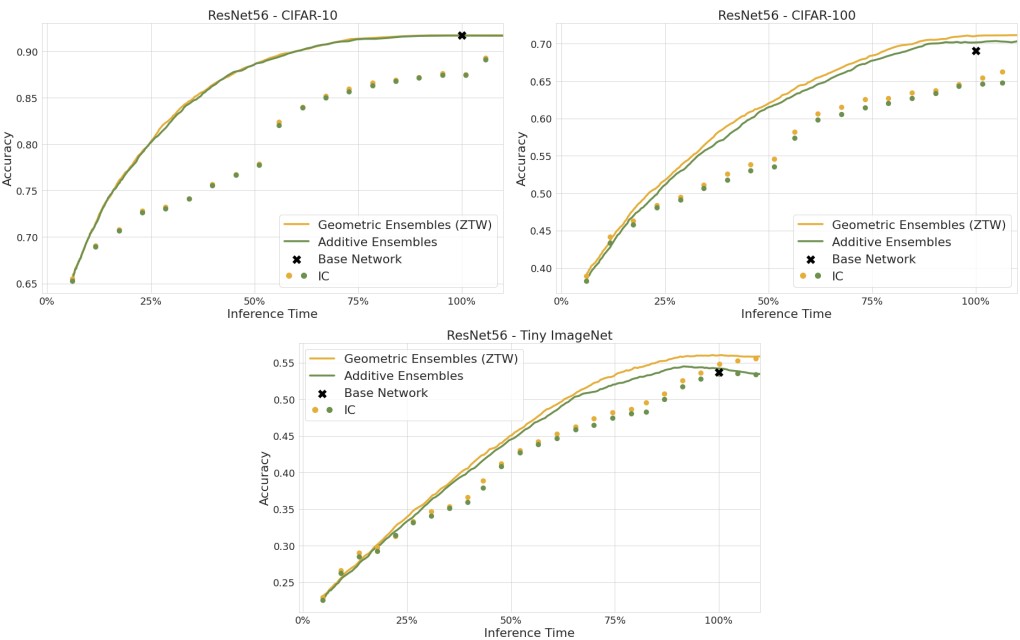

Figure 8: Comparison of geometric and additive ensembling on ResNet-56 with cascade connections, conducted on CIFAR-10, CIFAR-100, and Tiny ImageNet.

improving the first IC, using $\sum_k g_k$ will lead to a non-optimal update of its weights, thus reducing its predictive performance. With detach, $g_2 = g_3 = \ldots = 0$ and as such the cosine similarity is always $1$. This reasoning can be extended to the rest of ICs.

### C.4 Impact of the number of classes

Additionally, we check how the number of classes in the given problem impacts the results of each method. To do this, we take the CIFAR-10 dataset, which consists of 10 classes and divide the examples into two more general classes, which can be approximately described as modes of transportation (includes airplane, automobile, horse, ship, truck) and animals (bird, cat, deer, dog, frog). Thus, we obtain a dataset for binary classification which we dub CIFAR-2. We train and evaluate the proposed methods on this dataset with different backbones. Results, summed up in Table 5, show that although performance of ZTW is always on par or better than the baselines, the gap in performance is much smaller, with SDN achieving identical performance in some cases. Although, this might be due to the fact that CIFAR-2 is simpler than original CIFAR-10, we note that Zero Time Waste is better suited to non-binary classification problems.

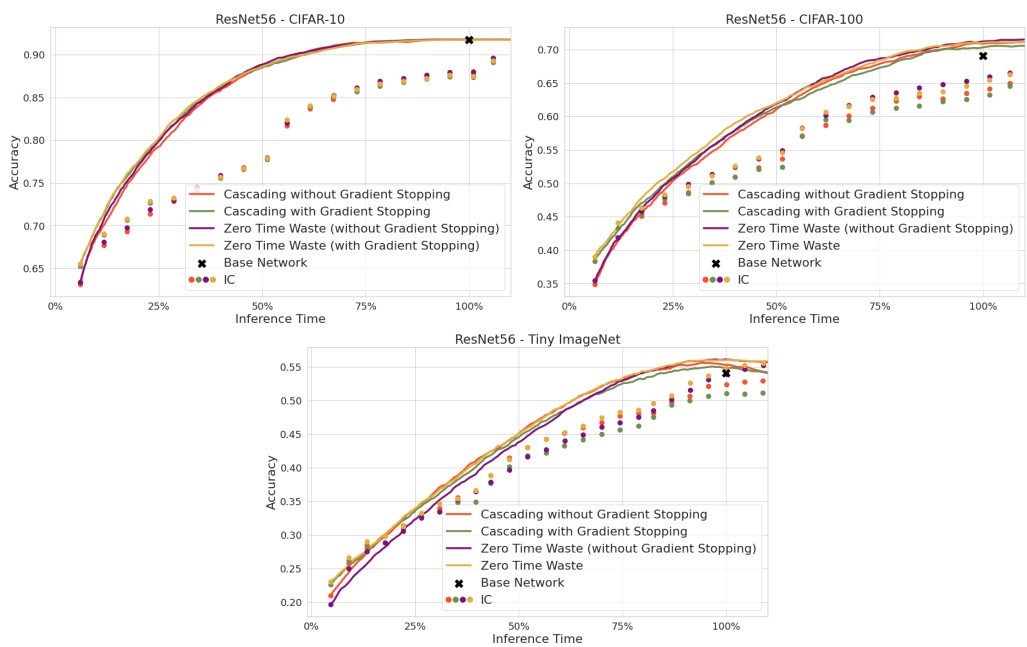

Figure 9: Effects of stopping gradient in ResNet-56 trained on CIFAR-10, CIFAR-100, and Tiny ImageNet.

Table 5: Results on the CIFAR-2 dataset.

| ResNet-56 | | | | | | |
|---|---|---|---|---|---|---|
| Data | Algo | 25% | 50% | 75% | 100% | Max |
| **ResNet-56** | SDN | 95.5 | 96.5 | 96.5 | 96.5 | 96.5 |
| | PBEE | 91.2 | 94.1 | 96.3 | 96.5 | 96.6 |
| | ZTW | 95.7 | 96.6 | 96.6 | 96.6 | 96.6 |
| **VGG** | SDN | 96.6 | 97.6 | 97.6 | 97.6 | 97.7 |
| | PBEE | 91.2 | 96.4 | 97.2 | 97.4 | 97.6 |
| | ZTW | 96.7 | 97.6 | 97.7 | 97.7 | 97.7 |
| **WideResNet** | SDN | 95.2 | 97.0 | 97.3 | 97.3 | 97.4 |
| | PBEE | 89.3 | 93.0 | 95.9 | 97.0 | 97.4 |
| | ZTW | 96.3 | 97.4 | 97.6 | 97.6 | 97.6 |
| **MobileNet** | SDN | 95.7 | 96.4 | 96.4 | 96.4 | 96.4 |
| | PBEE | 91.9 | 94.3 | 96.2 | 96.4 | 96.4 |
| | ZTW | 96.0 | 96.4 | 96.4 | 96.4 | 96.4 |

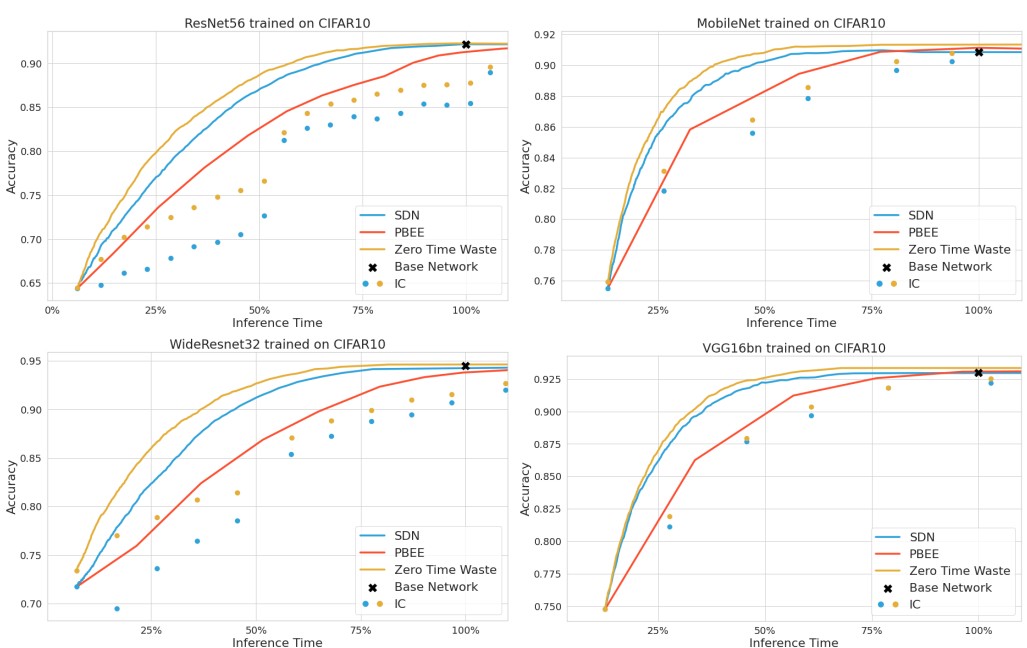

Figure 10: Inference time vs. accuracy obtained on various architectures trained on CIFAR-10.

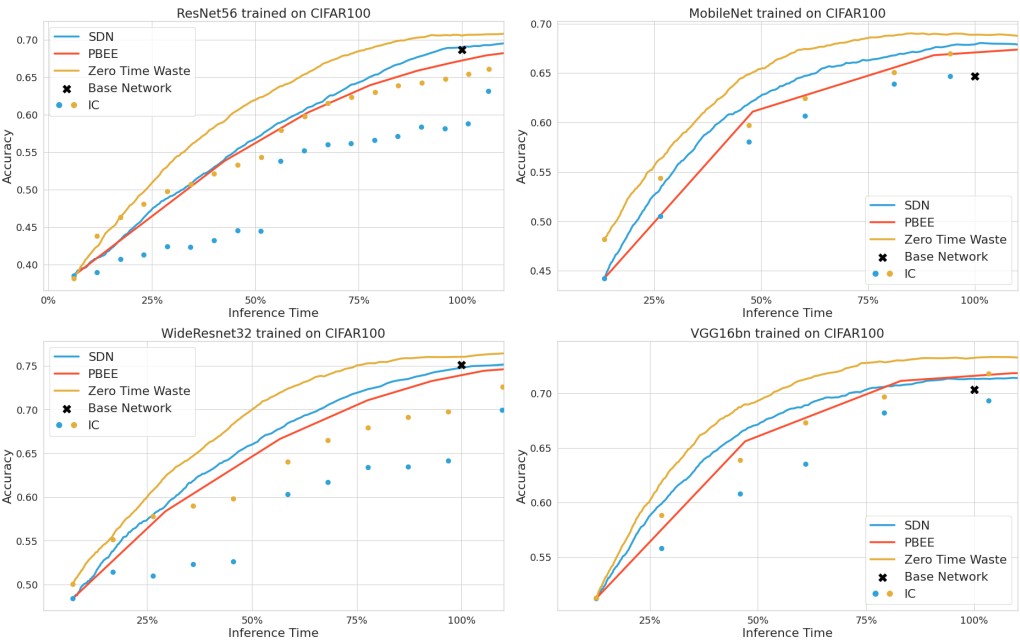

Figure 11: Inference time vs. accuracy obtained on various architectures trained on CIFAR-100.

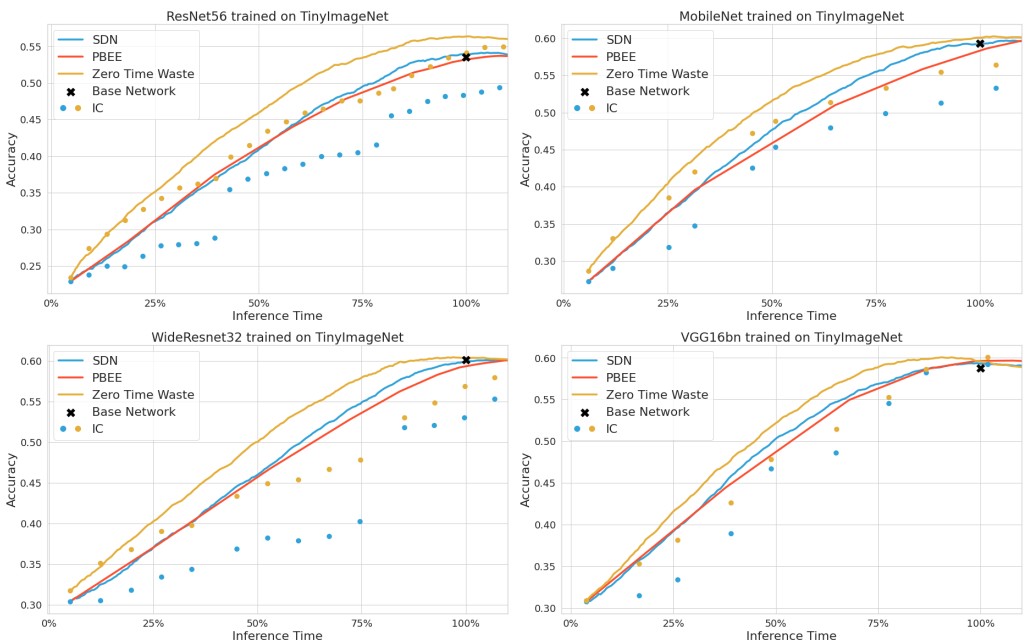

Figure 12: Inference time vs. accuracy obtained on various architectures trained on Tiny ImageNet.

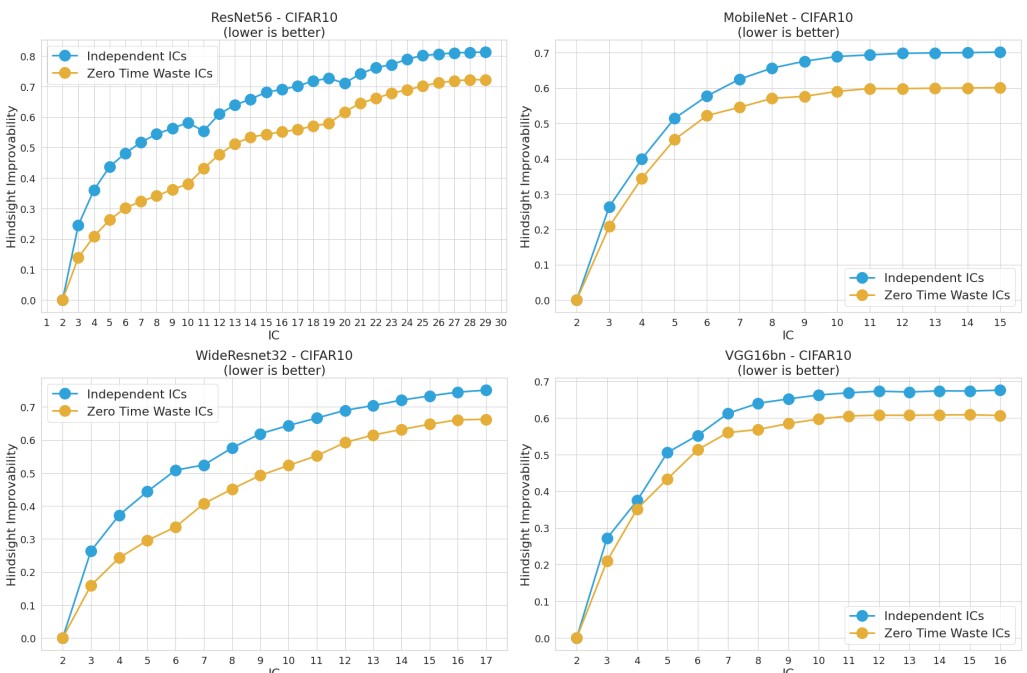

Figure 13: Hindsight Improvability of various architectures trained on CIFAR-10.

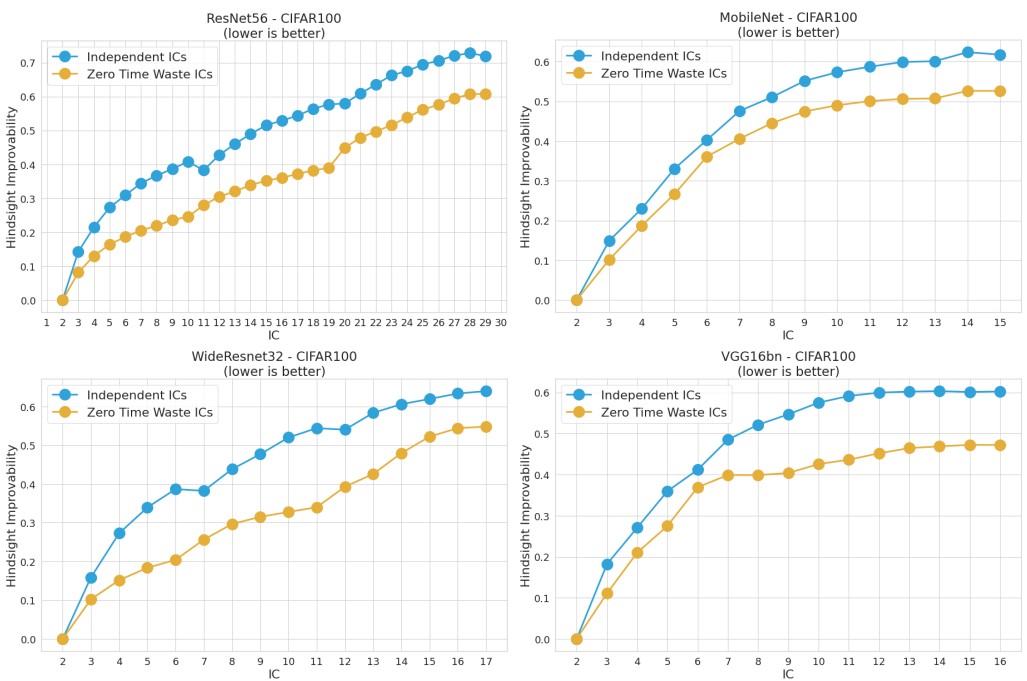

Figure 14: Hindsight Improvability of various architectures trained on CIFAR-100.

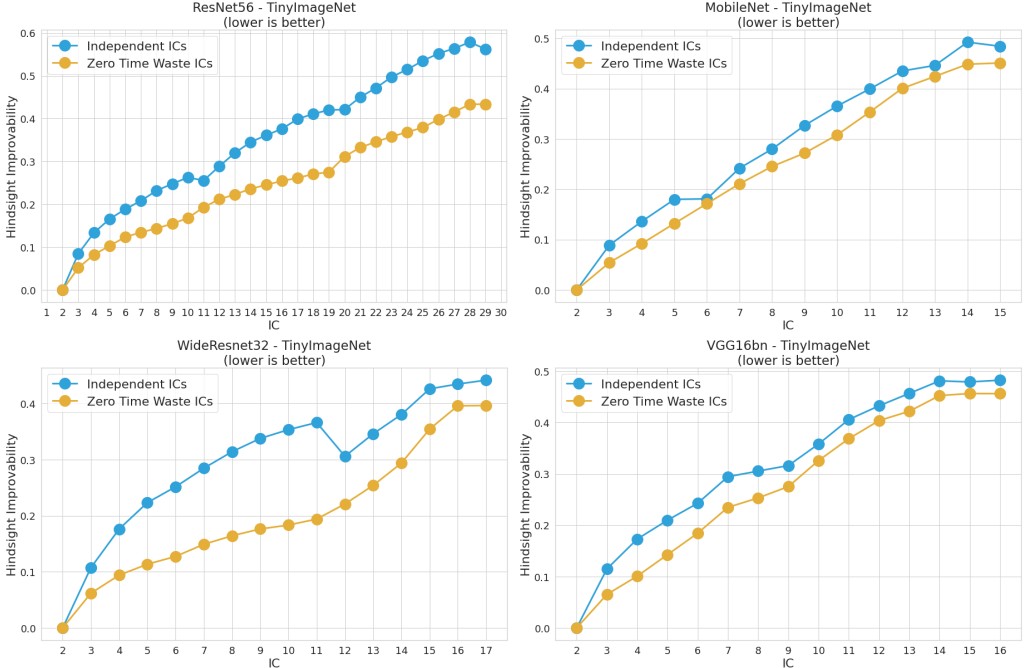

Figure 15: Hindsight Improvability of various architectures trained on Tiny ImageNet.