# OpenReview forum: "Zero Time Waste: Recycling Predictions in Early Exit Neural Networks"
_NeurIPS.cc/2021/Conference — NeurIPS 2021 Poster_

### Official Review · Reviewer_5x5Z · 2021-07-12

**Rating:** 4
**Confidence:** 4

**Summary:**

The paper propose a neural architecture with a backbone and side branches. These side branches leads to classification decisions. An ensemble method is use to combine the output of the side branches accumulatively (see Eq. 2). This is an interesting approach and its impact can go beyond early exit applications.

**Ethical Concerns:**

NIL

**Limitations And Societal Impact:**

Yes. This method may be useful if the network can be easily broken down into many subnetworks.

**Main Review:**

A major issue is with the understanding of Eq. (1) in this paper. Does g_\phi take in one argument of two arguments?

It would strengthen the paper if there is some theory showing the ensemble method performs better than non-ensemble method. Even if the author show very simple examples as illustrations, that would shed light in the understanding of this approach.

Will be good if the author can articulate other impactful contributions of this approach beyond the zero time waste selling point. My sense is that there may be other good properties of this approach.

**Time Spent Reviewing:**

2 hours

---

> ### Author Response · Authors · 2021-08-09
> **Authors' response to Reviewer 5x5Z**
>
> Thank you for your interest and, in particular, noticing that the impact of combining branches can “go beyond early exit applications”. We believe that ZTW is a promising approach.
>
> **Q1: A major issue is with the understanding of Eq. (1) in this paper. Does g_\phi take in one argument or two arguments?**
>
> Function $g_{\phi_m}()$ corresponds to running a neural network on the concatenation (in our setup “cascading”, see also Fig. 1) of the current m-th output of the backbone model denoted with $f_{\theta_m}$ with the previous $(m-1)$-th combination. As such, $g_\phi$ is a two-parameter function. We will improve the readability of Eq. (1).
>
> **Q2: It would strengthen the paper if there is some theory showing the ensemble method performs better than non-ensemble method. Even if the author shows very simple examples as illustrations, that would shed light on the understanding of this approach.**
>
> We have based our approach on the weak classifier theory [Schapire, Freund, 2012] and models like gradient boosting [Friedeman, 2001] (please see the related work subsection). They all show that a properly grouped ensemble of diverse weak (even only a bit better than random) results in a strong classifier.
>
> We have proposed a novel combination of classifier grouping composed of cascade connections that implement a linear forward knowledge transfer between successive ICs, and geometric ensembling which combines all previous ICs to provide a probabilistic prior for next IC to use (there is a vast literature on the beneficial impact of ensembles, see eg. [Hastie et al. 2008, Dietterich, 2002, Lakshminarayann et al, 2017], some cited in paper)
>
> Additionally, the proposed Hindsight Improvability measure shows clearly that combining ICs into ensembles gives advantages - see Fig. 2 for two independent examples. The combined output for an ensemble model (yellow dotted line) is much better than one built using independent ICs. In our opinion, the differences are huge.
>
> **Q3: Will be good if the author can articulate other impactful contributions of this approach beyond the zero time waste selling point. My sense is that there may be other good properties of this approach.**
>
> Thank you for your comment, as it is important. As the neural systems are becoming more complex, we have put an emphasis on time and energy waste as the positive feature of ZTW. We consider pointing out the wastefulness of current deep learning methods and proposing ways to solve it an important step towards positive societal impact and a crucial contribution of our work.
>
> An important feature of ZTW is the novel combination of the cascading with geometric ensembling towards a method that proves to be effective; the geometric ensembling emphasizes the relevance of an agreement between the ICs (see appendix C.2 for a discussion, also Fig. 2 for an example how ZTW might first be driven, and mistaken, by low level features, only to combine them using higher level ones); we believe that this proposition is an important one. Naturally the main aim is to speed up the system response in response time essential applications.
>
> We have applied the system in a reinforcement learning RL approach (see Experiments section 4.3) which, to our best knowledge, has not been done before, but is so important in domains such as autonomous cars, where fast processing of simple cases is of utmost importance (together with the energy saved),
>
> The proposed ZTW architecture is also fit for transfer learning, where first only the last feed-forward layer of the backbone network is updated to a new problem, while the added ICs, together with their cascading/ensembling connections, are trained on a new data set.
>
> **Limitations: This method may be useful if the network can be easily broken down into many subnetworks.**
>
> This was the assumption on the base system to be comparable. On the other hand, in the ZTW the backbone network layers can be selected freely with not all blocks used to build internal classifiers. We have not gone into details, but depending on the backbone network architecture it might either be better to select them evenly, or skip more layers near the input, whereas take more at the output (see e.g. [Scardapane et al, 2020]). This can save computations keeping the ZTW’s final accuracy. Finally, it would be interesting to apply ZTW to other types of architectures (e.g. recurrent networks, graph neural networks), but we argue this is out of scope of this work which aims to showcase the idea of reducing waste of neural networks.
>
> [Schapire, Freund, 2012] Schapire R. E., Freund Y., Boosting: foundations and algorithms, MIT Press (2012)
>
> [Friedman, 2001] Friedman J. H. Greedy function approximation: a gradient boosting machine, Ann. Stat 29(5), pp. 1189-1232 (2001)
>
> [Scardapane et al, 2020] Scardapane S., Scarpiniti M., Baccarelli E., Uncini A., Why should we add early exits to neural networks?, Cognitive Computation 12, pp. 954-966 (2020)
>
> [Hastie et al, 2008] Hastie T., Tibshirani R., Friedman R., The elements of statistical learning, 2nd ed., Springer (2008)
>
> [Dietterich 2002] Dietterich T., Ensemble learning, in The handbook of Brain theory and neural networks, ed. Arbib M., MIT (2002)
>
> [Lakshminarayann et al 2017] Laksminarayann B., Pritzel A., Blundell Ch. Simple and scalable predictive uncertainty estimation using deep ensembles, pp. 6402-6413, NeurIPS (2017)

---

> > ### Comment · Reviewer_5x5Z · 2021-09-10
> > **acknowledge the author's responses**
> >
> > I like to thank the authors for putting in a lot of hard work to produce this wonderful piece of work. Also thank for authors for carefully response to my reviews. Would be really helpful to see the improvements in the paper. Look forward to that.

---

### Official Review · Reviewer_etSV · 2021-07-15

**Rating:** 4
**Confidence:** 4

**Summary:**

This paper proposes a new approach, named Zero Time Waste (ZTW), which improves the performance of the neural networks with multiple exits by reusing the early generated predictions.  Extensive experiments across various datasets and architectures are conducted to demonstrate the effectiveness of the proposed ZTW.

**Ethics Review Area:**

["I don’t know"]

**Limitations And Societal Impact:**

Missing potential negative societal impact.

**Main Review:**

Strengths:
1. The paper is well written and the explanation of the proposed approach is easy to follow.
2. The analysis experiments in Section 4.2 are informative and interesting.
3. The application of early exit methods to reinforcement learning is new.

Weaknesses:
1. Overall, the novelty is limited. The proposed ZTW  reuses early predictions by:
    a. Adding direct connections between ICs. A similar reusing manner has been proposed in [1], where the proposed Inline Subnetwork Collaboration (ISC) transfers the early knowledge to later predictors.
    b. Combining previous outputs in an ensemble-like manner. The multi-exit network proposed in [2] also ensembles predictions of all completed exits during inference, which is similar to the proposed ZTW.
    The authors should discuss the difference between the ISC and their method.
2. The comparisons in Section 4.1 are insufficient. More networks with multi-exit architecture need to be evaluated to show the effectiveness of the proposed ZTW. The MSDNet [3] and its modification with forward-direct connections [1] are recommended.
3. What does the percentage in Table 1 mean? In line 224, it is said that the fraction is the computational cost measured by FLOPs. While, in line 226, it is said the 25% means the inference time. Note that the actual run time can be related but not limited to FLOPs, the authors should make it clear.
4. The experiments in Section 4.1 seem unfair. As it is stated in A.1, the core networks are trained as pre-trained models. Then the ICs of the proposed ZTW are further trained for 50 epochs. Moreover, the ensembling part of ZTW is trained for 500 epochs. How to guarantee that the performance gain comes from the ZTW rather than the addition training procedure.
5. Some related references are missing:
[1] Li H, Zhang H, Qi X, et al. Improved techniques for training adaptive deep networks. ICCV 2019.
[2] Phuong M, Lampert C H. Distillation-based training for multi-exit architectures, ICCV, 2019.
[3] Huang G, Chen D, Li T, et al. Multi-scale dense networks for resource efficient image classification. ICLR, 2018.

Additional comments:

1.What will the network performance be if all inputs exit from the same classifiers?

2.In [3], it is stated that attaching intermedia classifiers can affect the performance of the final predictors. Do similar phenomenons happen in ZTW?




**Time Spent Reviewing:**

24

---

> ### Author Response · Authors · 2021-08-09
> **Authors' response to Reviewer etSV**
>
> Thank you for the valuable and constructive criticism.
>
> **Q1 (Novelty), Q2 (Comparisons in Section 4.1) and Q5 (Missing references)**
>
> In our problem setup we assume that a pretrained network is used, and we do not modify the weights of the original network during training. We can thus apply our method to almost any network found in the literature. In contrast, the authors of MSDNet explicitly propose their own architecture specially tuned to the case of attaching Internal Classifiers, and training them along with the core network. Similarly, a crucial component of the method proposed in [1], the Gradient Equilibrium, explicitly alleviates problems coming from such training. In our setup these components do not make sense and therefore a fair comparison seems impossible.
>
> We agree that ISC is similar to our cascading connections. However, we argue that the proposed method is applied in a different setting, with frozen pre-trained networks, which offers different challenges than in the referenced work. In particular, we do not have to consider the impact of the gradient on the base network, but instead we need to increase the expressiveness of each ICs as much as possible, since they can't rely on changing the features of the base network. We also found that applying layer normalization after ICs helps stabilize the training, a fact that we mention in the appendix A.1. Additionally, the empirical results suggest that the considered methods lead to different conclusions. In particular, Section 4.3 of [1] states that "deeper layers typically benefit more from ISC" - in case of ZTW we observe an inverse phenomenon, where the early layers benefit the most from cascade connections, which leads us to believe that the setting is considerably different. Finally, the two methods differ in details - we use the logits of the previous IC and concatenate them with features that go into the last linear layer of the current IC. On the other hand, ISC transforms the logits from the previous classifier with a linear layer, increasing its size so that the input size into the last linear layer of the current IC is doubled. We thank the reviewer for pointing out the similarities, we will include the mentioned paper to the related work section and make differences clear.
>
> As for [2], we actually did include it in the related work section, line 96. There are two crucial differences to simple averaging of ICs’ answers: addition of trainable parameters and geometric ensembling. In our experience bare ensembling rarely provides any improvement, sometimes even hampering accuracy. The addition of learnable parameters allows the model to weigh each IC for each class. The geometric ensembling emphasizes the requirement of an agreement between ICs, and we show that it achieves superior results in C.2.
>
> **Q3: What does the percentage in Table 1 mean?**
>
> It is the fraction of total inference cost measured in floating point operations. Indeed, the word "time" may be misleading as it may be understood as the wall clock time. We will rephrase any such occurrences.
>
> **Q4: The experiments in Section 4.1 seem unfair.**
>
> The sequential multiple-phase training is arguably the standard approach. It is used, for example, in the SDN paper, as well as in [1] that you cited. We did train our own base networks for CIFAR-10, CIFAR-100 and Tinyimagenet, but we used the hyperparameters and setup details from the SDN paper. For Imagenet we used a pretrained network from the torchvision package.
>
> We verified that the networks are properly trained by increasing the number of epochs for training. The additional training time did not improve the performance neither in the case of the base network, nor in the case of internal classifiers.
>
> **Additional comment #1: What will the network performance be if all inputs exit from the same classifiers?**
>
> The appendix contains Figures 5, 10, 11, 12, where the accuracy scores for individual ICs are marked.
>
> **Additional comment #2: In [3], it is stated that attaching intermedia classifiers can affect the performance of the final predictors. Do similar phenomenons happen in ZTW?**
>
> The setting in [3] optimizes the weights of the core network. We, however, freeze the pretrained model when training ICs (along with the BN statistics buffer, which is an easy thing to overlook). The final prediction of the original classifier is thus unaffected by our method. Attaching additional ICs affects succeeding ICs because of cascading and ensembling, and as we show in C.1 the effect on performance is positive.
>
> **Societal impact**
>
> We emphasize that the description of the impact of our work is present in section 4.4, and it includes discussion about potential negative impact. It describes cases when early exit methods cause incorrect predictions for samples which are correctly classified by a standard model. To the best of our knowledge such limitations analysis for the early exit methods was not performed before.
>
> [1] Li H, Zhang H, Qi X, et al. Improved techniques for training adaptive deep networks. ICCV 2019.
>
> [2] Phuong M, Lampert C H. Distillation-based training for multi-exit architectures, ICCV, 2019.
>
> [3] Huang G, Chen D, Li T, et al. Multi-scale dense networks for resource efficient image classification. ICLR, 2018.

---

### Official Review · Reviewer_ZimZ · 2021-07-16

**Rating:** 4
**Confidence:** 5

**Summary:**

This paper focuses on improving the computational efficiency of multi-exit networks. The authors aim to reuse the computation performed in previous internal classiﬁers (ICs) at latter ICs. The proposed method, Zero Time Waste (ZTW), includes two main components: (1) adding direct connections between ICs and (2) involving the previous predictions in an ensemble-like manner. Some results on CIFAR, Tiny ImageNet, and Atari games are provided.

**Limitations And Societal Impact:**

The authors have included the discussions on the limitations and potential negative societal impact in the paper. Some future directions have been pointed out.

**Main Review:**

In general, I think that the problem studied in this paper, namely improving the efficiency of deep networks with multi-exit structure, is interesting and of practical significance. However, this paper may be weak in novelty. My concerns are listed below:

(1) The first component of ZTM (i.e., adding direct connections between ICs) has been proposed in [*1] as "Inline Subnetwork Collaboration (ISC)".

(2) I think that there may exist an overlap between the two components of ZTW. Ideally, if the learned knowledge is able to be effectively transferred to the following ICs by the direct connections, the ensemble mechanism might not be necessary.

(3) This paper is based on a naive multi-exit network architecture. Considering or at least discussing more SOTA networks (e.g., MSDNet, RANet, and GFNet) will be helpful to improve the quality.

(4) It is indeed novel to applying multi-exit networks to RL. However, in Figure 3, it seems that individually ICs outperform performing early exiting in many cases?



[*1] Li, H., Zhang, H., Qi, X., Yang, R., & Huang, G. (2019). Improved techniques for training adaptive deep networks. In Proceedings of the IEEE/CVF International Conference on Computer Vision (pp. 1891-1900).

**Time Spent Reviewing:**

3 hours

---

> ### Author Response · Authors · 2021-08-09
> **Authors' response to Reviewer ZimZ**
>
> Thank you for the review and valuable comments.
>
> **Q1: The first component of ZTM (i.e., adding direct connections between ICs) has been proposed in [1] as "Inline Subnetwork Collaboration (ISC)".**
>
> We are thankful for pointing out the paper concerning “Inline Subnetwork Collaboration (ISC)”. Indeed, ISC also uses direct connections between ICs, which we will mention in the camera-ready version. However, the authors of MSDNet explicitly propose their own architecture specially tuned to the case of attaching Internal Classifiers, and training them along with the core network. In contrast, our methodology shows how to efficiently attach internal classifiers to almost any pretrained network proposed in the literature. In consequence, making use of our approach one can speed up the inference time of the well-known network models such ResNet or VGG without modifying their weights.
>
> Our problem setup offers different challenges than in the referenced work. In particular, we do not have to consider the impact of the gradient on the base network, but instead we need to increase the expressiveness of each IC as much as possible, since they can't rely on changing the features of the base network. We also found that applying layer normalization after ICs helps stabilize the training, a fact that we mention in the appendix. Additionally, the empirical results suggest that the considered methods lead to different conclusions. In particular, Section 4.3 of [1] states that "deeper layers typically benefit more from ISC" - in case of ZTW we observe an inverse phenomenon, where the early layers benefit the most from cascade connections, which leads us to believe that the setting is considerably different. Finally, the two methods differ in details - we use the logits of the previous IC and concatenate them with features that go into the last linear layer of the current IC. On the other hand, ISC transforms the logits from the previous classifier with a linear layer, increasing its size so that the input size into the last linear layer of the current IC is doubled.
>
> **Q2: I think that there may exist an overlap between the two components of ZTW. Ideally, if the learned knowledge is able to be effectively transferred to the following ICs by the direct connections, the ensemble mechanism might not be necessary.**
>
> This aspect has been discussed and empirically evaluated in the Appendix, section C.1. The conclusion is that (lines 562-564): “combining these methods consistently improves the performance each of them achieved independently. Thus we argue that both cascade connections and geometric ensembling are required in Zero Time Waste and using only one of them will lead to significant performance deterioration.”
>
> **Q3: This paper is based on a naive multi-exit network architecture. Considering or at least discussing more SOTA networks (e.g., MSDNet, RANet, and GFNet) will be helpful to improve the quality.**
>
> Our paper addresses the problem of speeding up the inference time in pre-trained neural networks without modifying their weights or any prior knowledge about the architecture itself. In consequence, our methodology can be used as a universal plugin to existing models improving their inference time while saving most of their performance. We showed experimentally that our approach can be successfully used together with a broad range of modern architectures and benchmark datasets including ImageNet. This problem has been of interest in many recent works including [2, 3] and differs significantly from the one considered in reference papers. Taking this fact and the restrictive page limit into account, we did not elaborate on that topic in the related work section.
>
> **Q4: It is indeed novel to apply multi-exit networks to RL. However, in Figure 3, it seems that individually ICs outperform performing early exiting in many cases?**
>
> In the case of the reinforcement learning experiment, we used only two ICs because the architecture was much smaller than in the case of image data. In consequence, one can expect that the gain of using ensemble-like approaches should be lower. However, in the case of Qbert-v0, we save around 45% of computations without score degradation. For Pong we can get a 60% reduction with a minor impact on performance. As noticed by the Reviewer, the use of a single IC alone gives better results on average in some cases. Note, however, that the standard deviations in this environment are larger, and so the difference may not be significant. We present results for more environments in appendix B.3.
>
> [1] Li, H., Zhang, H., Qi, X., Yang, R., & Huang, G. (2019). Improved techniques for training adaptive deep networks. In Proceedings of the IEEE/CVF International Conference on Computer Vision (pp. 1891-1900).
>
> [2] Kaya, Yigitcan, Sanghyun Hong, and Tudor Dumitras. "Shallow-deep networks: Understanding and mitigating network overthinking." International Conference on Machine Learning. PMLR, 2019.
>
> [3] Scardapane, Simone, et al. "Why should we add early exits to neural networks?." Cognitive Computation 12.5 (2020): 954-966.

---

### Official Review · Reviewer_CbiF · 2021-07-18

**Rating:** 6
**Confidence:** 4

**Summary:**

This paper proposes to use all the outputs from the internal classifiers in an early-exit network. The paper presents two novel ways of making use of the previous prediction and concludes with the weighted geometric mean as a better choice empirically. By taking care of the details, i.e., stop gradient and handling numerical issues during training, the authors report superior results compared to early-exit networks that do not make use of previous classification results in the later classifiers. The paper is well-written with comprehensive empirical results on CIFAR, TinyImageNet with some ImageNet, and Atari.

**Limitations And Societal Impact:**

I do not think this paper has negative societal impact

**Main Review:**

## Strengths

- A novel, simple, and intuitive idea of improving early exit networks
- The simple idea is very effective when implemented correctly
- The paper is easy to follow, well-written, and inspiring
- Comprehensive empirical results with many networks and small datasets together with some on a large dataset. Moreover, this paper evaluates beyond commonly used supervised setting and perform analysis for reinforcement learning

## Weaknesses

- Why weighted geometric average is not clear. While it works great, it would be great to provide the motivation and analysis for the proposed modeling strategy
- ImageNet results are rather limited and it is not clear what is the model used for ImageNet. I assume it is ResNet-50. If so, it seems that other acceleration might be more preferable compared to early exits. Specifically, filter pruning can achieve 73% FLOP [1] and 55% FLOP [2] without degrading top-1 accuracy while this paper achieves 75% FLOP with a 6% top-1 accuracy drop. A similar conclusion can be derived from the ResNet56 on CIFAR results when cross comparing with the pruning literature [1]. This calls into question the relevance of early exits given the pruning literature. It would be great if the paper discusses this aspect.

[1] Chin, Ting-Wu, et al. "Towards Efficient Model Compression via Learned Global Ranking." CVPR 2020.

[2] Hou, Yuenan, et al. "Network Pruning via Resource Reallocation." arXiv preprint arXiv:2103.01847 (2021).

**Time Spent Reviewing:**

1.5

---

> ### Author Response · Authors · 2021-08-09
> **Authors' response to Reviewer CbiF**
>
> Thank you for the comments and interesting insights.
>
> **Q1: Why weighted geometric average is not clear. While it works great, it would be great to provide the motivation and analysis for the proposed modeling strategy**
>
> We appreciate the suggestion to highlight the reasons for using geometric ensembles. We believe that geometric ensembles outperform additive ensembles since they enforce the need for all classifier heads to agree on the final answer. In Appendix C.3 we discuss the difference between geometric and additive ensembles and in Figure 8 we present the empirical superiority of the geometric ensemble over an additive ensemble on ResNet-56.
>
> **Q2: ImageNet results are rather limited and it is not clear what is the model used for ImageNet.**
>
> We use ResNet-50 for ImageNet experiments - we will make this clear in the revised version of the paper. We appreciate the proposal to discuss pruning methods in our work. We agree that it would be a good idea to add information about improvements that can be achieved through pruning methods. Regarding the question of relevance of early-exit methods, we recognise pruning and early-exit methods as orthogonal methods that could be combined to leverage advantages of both.  Results on how pruning and early-exiting affect time-vs-accuracy tradeoff separately and in combination would be very interesting. However, this is a general question that could be asked for any early-exit method, not only to our proposed ZTW, therefore we consider this exploration as outside the scope of our work.

---

### Decision · Program_Chairs · 2021-09-28

**Decision:**

Accept (Poster)

**Comment:**

This paper introduces an architecture of using all the outputs from the internal classifiers in an early-exit network. It is claimed the proposed method can achieve both more accurate results and faster prediction. Initially this paper receives 3 negative and 1 positive scores. The main problem of negative reviews concerns the novelty of the proposed method. The main trick of adding direct connections between ICs is very similar to what has been proposed in previous work of "Improved techniques for training adaptive deep network" by Zhang et al., although the setting is claimed to be different. After discussion, the positive reviewer realizes he/she has missed this reference, and decreases the original score. A more clear discussion of the differences between the existing technique and the proposed one should be conducted in the revision. I personally also think testing the proposed method on larger scale datasets such as ImageNet would be beneficial. I suggest the authors revise the paper according the the detailed reviews, which I believe would make it much stronger for future submissions.

**Consistency Experiment:**

NeurIPS has a long history of experimentation. In 2014, NeurIPS ran an experiment in which 10% of submissions were reviewed by two independent committees to quantify the randomness in the review process. This year, we repeated a variant of this experiment to see how the quality of the review process has changed over time.  This paper was part of the experiment and was therefore assigned to two committees (consisting of reviewers, an Area Chair, and a Senior Area Chair) that reached independent decisions.  If both committees made the same recommendation, this recommendation was followed. If a single committee recommended acceptance, the paper was accepted (with the exception of a few cases in which the other committee identified what we considered a fatal flaw, e.g., an error in a key result).

This copy’s committee reached the following decision: **Reject**

The other committee assigned to the paper recommended **Accept (Poster)**.  You can find the other set of reviews, along with any follow up discussion with the authors here:
https://openreview.net/forum?id=7AiFm-cB-ac